# LADA: Look-Ahead Data Acquisition via Augmentation for Deep Active Learning

**Yoon-Yeong Kim**[1]    **Kyungwoo Song**[2]    **Joonho Jang**[1]    **Il-Chul Moon**[1,3]
[1]Korea Advanced Institute of Science and Technology (KAIST)   [2]University of Seoul   [3]Summary.AI
yoonyeong.kim@kaist.ac.kr kyungwoo.song@uos.ac.kr
adkto8093@kaist.ac.kr icmoon@kaist.ac.kr

## Abstract

*Active learning* effectively collects data instances for training deep learning models when the labeled dataset is limited and the annotation cost is high. *Data augmentation* is another effective technique to enlarge the limited amount of labeled instances. The scarcity of labeled dataset leads us to consider the integration of data augmentation and active learning. One possible approach is a pipelined combination, which selects informative instances via the acquisition function and generates virtual instances from the selected instances via augmentation. However, this pipelined approach would not guarantee the informativeness of the virtual instances. This paper proposes Look-Ahead Data Acquisition via augmentation, or LADA framework, that looks ahead the effect of data augmentation in the process of acquisition. LADA jointly considers both 1) unlabeled data instance to be selected and 2) virtual data instance to be generated by data augmentation, to construct the acquisition function. Moreover, to generate maximally informative virtual instances, LADA optimizes the data augmentation policy to maximize the predictive acquisition score, resulting in the proposal of *InfoSTN* and *InfoMixup*. The experimental results of LADA show a significant improvement over the recent augmentation and acquisition baselines that were independently applied.

## 1 Introduction

Large-scale datasets in the big data era have opened the blooming of data science, but the data labeling requires significant efforts from human annotators, or *oracle*. Therefore, an adaptive sampling by an acquisition function, i.e. *active learning*, has been developed to select the most informative data instances in learning the decision boundary [1–3]. This selection is difficult because it is influenced by the learner and the dataset at the same time. Hence, the understanding of the relation between the two has become the components of *active learning*, which queries the next training example by the informativeness, assessed by acquisition function.

Besides *active learning*, *data augmentation* is another data source for learning models that provides virtual data instances generated from the training dataset [4]. Conventional data augmentation has been a simple transformation of labeled data instances, e.g., flipping, rotating, etc [5]. Recently, the data augmentation has expanded to become a deep neural model generating virtual instances, such as Generative Adversarial Networks (GAN) [6] or Variational Autoencoder (VAE) [7]. Spatial Transform Networks (STN) [8] also generate spatially transformed instances for learning the classifier. Since the conventional and the deep neural model-based augmentations perform the Vicinal Risk Minimization (VRM) [9], they preserve labels of virtual instances and limit the feasible vicinity. To overcome the limited vicinity of VRM, *Mixup* [10] and its variants have been proposed by interpolating multiple data instances. The pair of interpolated features and labels, or the *Mixup* instances, become virtual instances to enlarge the support of the data distribution.

35th Conference on Neural Information Processing Systems (NeurIPS 2021).

Given the scarce labeled dataset, it is natural to consider combining *active learning* and *data augmentation*. One possible way is a pipelined approach, which selects data instances by an acquisition function and generates virtual instances from the selected instances by an augmentation model afterward [11]. However, the acquisition function does not consider the potential gain from the augmentation in the assessment of the informativeness. Hence, without any feedback or integration effort at the acquisition level, the virtual instances generated by data augmentation would not guarantee the informativeness. Figure 1a illustrates the pipelined combination, where the averaged entropy value of the virtual instances is 1.61.

This paper proposes the Look-Ahead Data Acquisition via augmentation, or LADA framework. LADA looks ahead the effect of data augmentation in advance of the actual acquisition process, by selecting data instances according to the acquisition score of both unlabeled real instances and their augmented virtual instances, at the same time. The acquisition algorithm of LADA enables us to train the classifier with the instances that are informative 1) when labeled by *oracle* and 2) when augmented via data augmentation. Furthermore, the data augmentation policy in LADA is trained to maximize the acquisition score of the virtual instances. Figure 1b illustrates the different behavior of LADA with *Max Entropy* and *Mixup*, where the averaged entropy value of the virtual instances is 2.12, which is higher than the pipelined combination in Figure 1a.

Here are our contributions. First, we propose a generalized framework, named LADA, that looks ahead the acquisition score of the virtual data instances to be augmented, in advance of the acquisition. Second, we train the data augmentation policy to maximize the acquisition score, hence generate informative virtual instances. Particularly, we propose two data augmentation methods, *InfoSTN* and *InfoMixup*, which are trained by the feedback of acquisition scores. Third, we instantiate the proposed framework with various combinations of acquisition-augmentation of known methods. There have been some prior works that suggest the concept of look-ahead, without acknowledging the value of augmentation [12, 3, 13]. We claim our novelty for look-ahead in conjunction with the augmentation of virtual instances. Moreover, look-ahead is a necessary concept in any active learning scheme because the active learning requires an active seeking on high-value data instances which will impact the classifier if they are used in the inference, so this assessment on the impact becomes the look-ahead in such active learning concept.

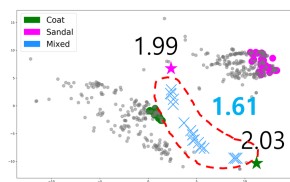

(a) Pipelined combination of *Max Entropy* and *Mixup*

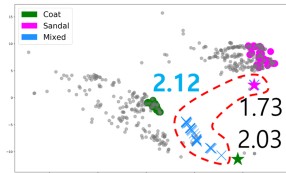

(b) LADA with *Max Entropy* and *Mixup*

Figure 1: Illustration of different behaviors of the acquisition process in active learning; with selected instances ($\star$), virtual instances ($\times$), entropy values of selected instances (black number), and averaged entropy values of virtual instances (blue number).

## 2 Preliminaries

### 2.1 Problem Formulation

This paper trains a classifier network parameterized by $\theta$, i.e., $f_\theta$, with dataset $\mathscr{X}$ while our scenario is differentiated by assuming $\mathscr{X} = \mathscr{X}_U \cup \mathscr{X}_L$ and $|\mathscr{X}_U| \gg |\mathscr{X}_L|$. Here, $\mathscr{X}_U$ is a set of unlabeled data instances, and $\mathscr{X}_L$ is a set of labeled data instances. Given these notations, a data augmentation function, $f_{aug}(x; \tau)\colon \mathscr{X} \to \mathscr{T}(\mathscr{X})$, transforms a data, $x \in \mathscr{X}$, into a modified data, $\tilde{x} \in \mathscr{T}(\mathscr{X})$; where $\tau$ is a parameter describing the policy of transformation, and $\mathscr{T}(\mathscr{X})$ is the transformed set of $\mathscr{X}$. On the other hand, a data acquisition function, $f_{acq}(x; f_\theta)\colon \mathscr{X}_U \to \mathbb{R}$, calculates a score of each data instance, $x \in \mathscr{X}_U$, based on the current classifier, $f_\theta$. $f_{acq}$ provides the criteria of selection strategy in the learning procedure of $f_\theta$ with the instance, $x \in \mathscr{X}_U$. We categorize the acquisition functions and the augmentation functions by placing the name of the algorithm as superscript.

### 2.2 Data Augmentation

In the conventional data augmentations, $\tau$ in $f_{aug}(x; \tau)$ indicates the predefined degree of rotating, flipping, cropping, etc. $\tau$ is manually chosen to describe the vicinity of each data instance.

Another approach of modeling $\tau$ is utilizing the feedback from the current classifier network, $f_\theta$. Spatial Transformer Network (STN) transforms a data using a grid sampler generated through a

localisation network parameterized by $\tau$, or $f_\tau$ [8]. The augmentation policy, $\tau$, in STN is trained by the cross-entropy (CE) loss of the transformed data with the ground-truth label, $y$, resulting in $\tau^* = \text{argmin}_\tau CE(f_\theta(f_{aug}^{STN}(x; \tau)), y)$.

In a recent work, *Mixup*-based data augmentations generate a virtual data instance in between a pair of data instances to overcome the limited vicinity. In *Mixup*, $\tau$ becomes the mixing policy of two data instances, $x_i$ and $x_j$, as $f_{aug}^{Mixup}(x_i, x_j; \tau) = \lambda x_i + (1 - \lambda)x_j, \lambda \sim \text{Beta}(\tau, \tau)$, where the labels are also mixed by the proportion $\lambda$ [10]. Further from mixing input features, *Manifold Mixup* mixes the hidden features from the multiple middle layers of neural networks to learn a smoother decision boundary [14]. Whereas $\tau$ is a fixed value without learning process so far, *AdaMixup* learns $\tau$ by adopting a discriminator, $\varphi^{ada}$, as $\tau^* = \text{argmax}_\tau[\log \text{P}(\varphi^{ada}(f_{aug}^{Mixup}(x_i, x_j; \tau)) = 1) + \log \text{P}(\varphi^{ada}(x_i) = 0) + \log \text{P}(\varphi^{ada}(x_j) = 0)]$ [15].

### 2.3 Active Learning

We focus on the pool-based active learning with an uncertainty score [3]. Given this scope of active learning, the data acquisition function measures the utility score of the unlabeled data instances, i.e. $x_U^* = \text{argmax}_{x_U \in \mathscr{X}_U} f_{acq}(x; f_\theta)$. The traditional acquisition functions measure the predictive entropy, $f_{acq}^{Ent}(x_U; f_\theta) = \mathbb{H}[\hat{y}|x_U; f_\theta]$ [16], where $\mathbb{H} := -\sum_c \text{P}(\hat{y} = c|x_U; f_\theta) \log_2 \text{P}(\hat{y} = c|x_U; f_\theta)$ and $\hat{y} = f_\theta(x_U)$; or the variation ratio, $f_{acq}^{Var}(x_U; f_\theta) = 1 - max_{\hat{y}}\text{P}(\hat{y}|x_U; f_\theta)$ [17]. Bayesian approaches for active learning are also proposed as $f_{acq}^{BALD}(x_U; f_\theta) = \mathbb{H}[\hat{y}|x_U; f_\theta] - \mathbb{E}_{\text{P}(\theta|D_{train})}[\mathbb{H}[\hat{y}|x_U; f_\theta]]$ [18].

Additional modules are also applied to measure the acquisition score. Variational Adversarial Active Learning (VAAL) introduces a discriminator, $\varphi^{VAAL}$, to estimate the probability of $x_U$ belonging to $\mathscr{X}_U$, as $f_{acq}^{VAAL}(x_U; \varphi^{VAAL}) = \text{P}(x_U \in \mathscr{X}_U; \varphi^{VAAL})$ [19]. Learning Loss for Active Learning (LL4AL) adopts a simple neural network called a loss prediction module, or $f_{LPM}$ [20]. $f_{LPM}$ is trained to predict the loss of each data, and the instance with the highest predictive loss is selected, as $f_{acq}^{LL4AL}(x_U; f_{LPM}) = f_{LPM}(f_\theta^k(x_U)|k \in K)$. Here, $f_\theta^k(x_U)$ represents the $k$th hidden representation of $x_U$, and $K$ represents the set of hidden layers of the classifier, $f_\theta$.

### 2.4 Active Learning with Data Augmentation

There have been prior researches on leveraging data augmentation for active learning. Bayesian Generative Active Deep Learning (BGADL) combines acquisition and augmentation in a pipelined approach [11]; BGADL selects data instances via $f_{acq}$, and BGADL augments the selected instances via $f_{aug}$, which is VAE-ACGAN. However, BGADL limits the vicinity to preserve the label validity. Also, a large number of labeled instances are demanded to train the generative model, VAE-ACGAN, of BGADL at every acquisition round. More importantly, BGADL does not consider the potential gain from data augmentation in the process of acquisition.

In comparison with BGADL, Consistency-based Active Learning (CAL) algorithms consider data augmentation in the acquisition process, by replacing the uncertainty with augmentation-based inconsistency, resulting in $f_{acq}^{CAL}(x; f_\theta) = D[P(\hat{Y}|x, f_\theta), P(\hat{Y}|\tilde{x}, f_\theta)]$ [21]. Here, $D$ denotes the L2 norm [22] or KL divergence [23] that represents the inconsistency of the predictions from the transformation of the data instance $x$ into $\tilde{x}$. The algorithm in [21] selects a data instance that has the highest variance of class-wise predictions when it is transformed over a random set of data augmentations. However, the augmentation in [21] is not learnable, i.e., not optimized to enhance the informativeness of the augmented instance. Also, the augmentation is restricted to label-preserving transformations, such as random cropping or horizontal flipping, to measure the dissimilarity in predictions when the input is perturbed with the perceptual content of the instance being preserved.

## 3 Methodology

This paper differentiates itself from the previous acquisition-augmentation integration by presenting the learnable augmentations in conjunction with the potential acquisition scores of the virtual instances. Therefore, we start by formulating such a learnable framework in Section 3.1. Afterward, we propose an integrated function for acquisition and augmentation as the implementation of the framework in Section 3.2 and Section 3.3. Particularly, we propose adaptive versions of augmentation, i.e.,

*InfoSTN* and *InfoMixup*, whose policies are learned by the feedback of data acquisition score, i.e. *Max Entropy*. It should be noted that *InfoMixup*, as well as *InfoSTN*, can adopt various types of acquisition functions, other than *Max Entropy*, for feedback to train, see Appendix D and Section 4.

## 3.1 Look-Ahead Data Acquisition via Augmentation

Since we look ahead the acquisition score of the augmented data instances, it is natural to integrate the functionalities of acquisitions and augmentations. This paper proposes a framework of Look-Ahead Data Acquisition via augmentation, or the LADA framework, see Figure 2. The goal of LADA is to enhance the informativeness of both 1) real-world data instance, which is unlabeled at current, but will be labeled by the *oracle* in the future; and 2) virtual data instance, which will be generated from the unlabeled data instances

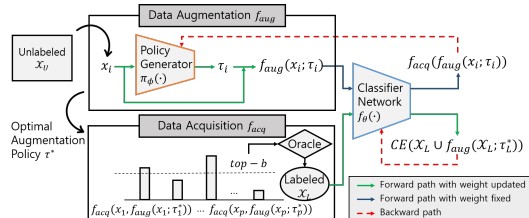

Figure 2: Overview of LADA framework

that are selected. This goal is achieved by looking ahead of the virtual examples' acquisition scores before actual selection.

Specifically, LADA trains the augmentation policy, $\tau$, of $f_{aug}(x; \tau)$ to maximize the acquisition score of the transformed data instance of $x_U$ before the *oracle* annotations. Eq. 1 specifies the learning objectives of the augmentation policy via the feedback from acquisition.

$$\tau^* = \operatorname*{argmax}_{\tau} f_{acq}(f_{aug}(x_U; \tau); f_\theta) \tag{1}$$

With the optimal $\tau^*$ corresponding to $x_U$, LADA calculates the acquisition score of $x_U$ for selection, by considering the utility of both $x_U$ and their augmented instances, $f_{aug}(x_U; \tau^*)$, as Eq. 2. In Eq. 2, $\gamma$ weights the relative importance of the acquisition from the virtual instance.

$$x_U^* = \operatorname*{argmax}_{x_U \in \mathscr{X}_U} [f_{acq}(x_U; f_\theta) + \gamma f_{acq}(f_{aug}(x_U; \tau^*); f_\theta)] \tag{2}$$

To begin with, we introduce an integrated single function to substitute the composition of functions as $f_{integ} = f_{acq} \circ f_{aug}(x_U) = f_{acq}(f_{aug}(x_U; \tau); f_\theta)$ for generality. $f_{integ}$ is a general formalism for LADA that 1) constructs a part of the acquisition function that looks ahead the informativeness of the virtual data instances, as Eq. 2. Also, $f_{integ}$ 2) becomes the objective function for training the augmentation policy to maximize the informativeness of the virtual instances, as Eq. 1.

If we choose the simplest form of LADA, $f_{integ}$ can be a simple composition of well-known acquisition functions and augmentation functions where the policy of augmentation is fixed. However, this does not generate maximally informative virtual data instances. Hence, we propose the integration where the policy of data augmentation is trained to maximize the acquisition score, within a single function. We name the fixed integration case as LADA$^{\text{fixed}}$, and compare it with LADA in Section 4.

## 3.2 Integrated Augmentation and Acquisition: *InfoSTN*

This section introduces LADA that adopts 1) STN for $f_{aug}$, i.e., $f_{aug}^{STN}$; and 2) *Max Entropy* for $f_{acq}$, i.e., $f_{acq}^{Ent}$, to instantiate a case of $f_{integ}$ as $f_{integ}^{InfoSTN}$, resulting in the proposal of *InfoSTN*.

### 3.2.1 Data Augmentation Policy Learning

STN is a learnable neural network inserted into the classifier network, $f_\theta$, which spatially manipulates the data instance, $x$ [8]. STN consists of three parts (see Appendix E); 1) The localization network, $f_\tau$, i.e., a neural network parameterized by $\tau$, regresses the transformation parameters, $\nu$. 2) The grid generator function, $f_T$, generates a grid, $g$, from a regular grid, $G$, using the transformation parameters, $\nu$. 3) Finally, the sampler function, $f_S$, is applied to the input data instance, $x$, with the generated grid, $g$, to get a transformed instance, $\tilde{x}$. The overall process sums up to $f_{aug}^{STN}(x; \tau) = f_S(x, g) = f_S(x, f_T(G; \nu)) = f_S(x, f_T(G; f_\tau(x)))$. Hence, the parameters of the localization network, or $\tau$, correspond to the augmentation policy of STN.

For $x_U \in \mathscr{X}_U$, we propose the adaptive version of STN, or *InfoSTN*, which is trained via the feedback from the acquisition function of active learning. *InfoSTN* learns its policy, $\tau$, with the objective

---

**Algorithm 1** LADA with *Max Entropy* and *Manifold Mixup*

---

**Input:** Labeled dataset $\mathscr{X}_L^0$, Classifier $f_\theta$

 1: **for** $j = 0, 1, 2, \ldots$ **do**                                                     ▷ active learning iteration
 2:      Get $\mathscr{X}_U'$ which is randomly shuffled $\mathscr{X}_U$
 3:      Randomly chose the layer index $k$ of the $f_\theta$
 4:      $\phi^* = \mathrm{argmin}_\phi \frac{1}{|\mathscr{X}_U|} \sum_{(x_i, x_i') \in (\mathscr{X}_U, \mathscr{X}_U')} L_\pi([h^k(x_i), h^k(x_i')])$         ▷ learning policy
 5:      $\tau_i^* = \pi_{\phi^*}([h^k(x_i), h^k(x_i')])$ for $(x_i, x_i') \in (\mathscr{X}_U, \mathscr{X}_U')$
 6:      Construct $f_{acq}^{LADA}$ as Eq. 13 and select and query the dataset, $\mathscr{X}_S$
 7:      Update the labeled dataset, $\mathscr{X}_L^{j+1} = \mathscr{X}_L^j \cup \mathscr{X}_S$
 8:      **for** $t = 0, 1, 2, \ldots$ **do**                                                    ▷ training $f_\theta$
 9:           Get virtual dataset, $\mathscr{X}_M$, from $\mathscr{X}_S$ using the optimal augmentation policy, $\tau_i^*$
10:           Update $\theta$ with the loss, $L_f$, as Eq. 14
11:      **end for**
12: **end for**

---

function of $f_{integ}^{InfoSTN}$, described as below:

$$f_{integ}^{InfoSTN}(x_U; \tau, f_\theta) = f_{acq}^{Ent}(f_{aug}^{STN}(x_U; \tau); f_\theta) = \mathbb{H}[\hat{y}|f_S(x_U, f_T(G; f_\tau(x_U))); f_\theta]. \quad (3)$$

Then, the augmentation policy, $\tau$, of *InfoSTN* in LADA is optimized to maximize the informativeness of the transformations of the unlabeled data instances, i.e., $f_{integ}^{InfoSTN}$, as below:

$$\tau^* = \mathrm{argmax}_\tau \frac{1}{|\mathscr{X}_U|} \sum_{x_U \in \mathscr{X}_U} \mathbb{H}[\hat{y}|f_S(x_U, f_T(G; f_\tau(x_U))); f_\theta]. \quad (4)$$

It should be noted that $\tau$ is originally designed to minimize the cross-entropy loss of the labeled data instances, at the training process of the classifier network, $f_\theta$. This optimization is in a different direction from the optimization of Eq. 4. The original optimization on $\tau$ is dedicated to exploiting the classifier, but Eq. 4 has an optimization component to explore the augmentation space. Hence, at the beginning of each acquisition iteration, we save the current parameters, $\tau$, of the localization network. Then, we load the saved parameters to insert to $f_\theta$ when learning the classifier $f_\theta$.

### 3.2.2 Acquisition Function by Learned Policy and Model Training

With the optimal policy, $\tau^*$, we construct the acquisition function, $f_{acq}^{LADA}$, that looks ahead the informativeness of 1) the unlabeled instances and 2) the transformed instance by *InfoSTN* with the optimal policy, $\tau^*$.

$$f_{acq}^{LADA}(x_U; f_\theta) = \mathbb{H}[\hat{y}|x_U; f_\theta] + \gamma \mathbb{H}[\hat{y}|f_S(x_U, f_T(G; f_{\tau^*}(x_U))); f_\theta] \quad (5)$$

For the active learning with the allowed budget per acquisition as $b$, we acquire the top-$b$ instances, i.e., $\mathscr{X}_S$, among the subsets, $\mathscr{X}_S' \subset \mathscr{X}_U$ with $|\mathscr{X}_S'| = b$, by the acquisition function, $f_{acq}^{LADA}$; $\mathscr{X}_S = \mathrm{argmax}_{\mathscr{X}_S' \subset \mathscr{X}_U} \sum_{x_i \in \mathscr{X}_S'} f_{acq}^{LADA}(x_i; f_\theta)$. After acquiring and labeling $\mathscr{X}_S$, we load the saved parameters, $\tau$, to current STN and insert the STN to the classifier network, $f_\theta$, for training with the labeled data instances and the augmented instances in an end-to-end fashion, see Appendix E.

### 3.3 Integrated Augmentation and Acquisition: *InfoMixup*

STN and our proposed variant, *InfoSTN*, are label-preserving data augmentations, which limit the vicinity of the transformed instances. Hence, to overcome the limitation of the vicinity, this section introduces LADA that adopts 1) *Mixup* for $f_{aug}$, i.e. $f_{aug}^{Mixup}$; and 2) *Max Entropy* for $f_{acq}$, i.e. $f_{acq}^{Ent}$, to instantiate another case of $f_{integ}$ as $f_{integ}^{InfoMixup}$, resulting in the proposal of *InfoMixup*. Here, we adopt *ManifoldMixup* to learn smoother decision boundary at multiple levels of representations.

### 3.3.1 Data Augmentation Policy Learning

*InfoMixup* is an adaptive version of *Mixup* to train the data augmentation via the feedback from an acquisition function. *InfoMixup* learns its mixing policy, $\tau_i$, corresponding to the $i$-th pair,

$(x_i, x_i') \in \mathscr{X}_U \times \mathscr{X}_U$, with the objective function of $f_{integ}^{InfoMixup}$ as Eq. 6.

$$f_{integ}^{InfoMixup}(x_i, x_i'; \tau_i, f_\theta) = f_{acq}^{Ent}(f_{aug}^{Mixup}(x_i, x_i'; \tau_i); f_\theta) \tag{6}$$

Then, the optimal mixing policy, $\tau_i^*$, of *InfoMixup* is found by the optimization as below:

$$\tau_i^* = \underset{\tau_i}{\mathrm{argmax}} f_{integ}^{InfoMixup}(x_i, x_i'; \tau_i, f_\theta). \tag{7}$$

Different from the learning process of $\tau$ in *InfoSTN*, we need a policy generator network, $\pi_\phi$, to perform an amortized inference on the Beta distribution of *InfoMixup*. While we provide the details in Section 4.1 and Figure 3, we formulate this inference process as Eq. 8 and Eq. 9[1].

$$h^k(x_i) = f_\theta^{0:k}(x_i), \quad h^k(x_i') = f_\theta^{0:k}(x_i') \tag{8}$$

$$\tau_i = \pi_\phi([h^k(x_i), h^k(x_i')]) = NN_\phi([h^k(x_i), h^k(x_i')]) \tag{9}$$

To train the parameters, $\phi$, of the policy generator network, $\pi_\phi$; the paired latent features, $h^k(x_i)$ and $h^k(x_i')$, are mixed-up with $N$ number of $\lambda_i^n \sim$ Beta$(\tau_i, \tau_i)$ to produce $h_{mix}^{k,n}(x_i, x_i'; \tau_i)$ as below:

$$h_{mix}^{k,n}(x_i, x_i'; \tau_i) = \lambda_i^n h^k(x_i) + (1 - \lambda_i^n)h^k(x_i'),$$
$$n \in \{1, \ldots, N\}. \tag{10}$$

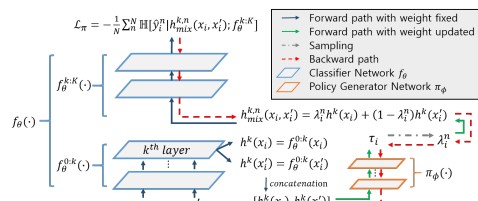

Figure 3: Training process of the policy generator network, $\pi_\phi$, in LADA with *Max Entropy* and *Manifold Mixup*

By processing $h_{mix}^{k,n}$ for the rest layers of the classifier network, the predictive class probability of the mixed features is obtained as $\hat{y}_i^n = f_\theta^{k:K}(h_{mix}^{k,n}(x_i, x_i'; \tau_i))$. Then, the policy generator network, $\pi_\phi$, is trained to minimize a loss function of Eq. 11, which is the negative value of the predictive entropy, so that the policy generates high entropy valued, or informative, virtual instances. The gradient of this loss function is calculated by averaging the $N$ entropy values of the replicated mixed features.

$$\frac{\partial}{\partial\phi}L_\pi([h^k(x_i), h^k(x_i')]) = \frac{\partial}{\partial\phi}(-\frac{1}{N}\sum_{n=1}^{N}\mathbb{H}[\hat{y}_i^n | h_{mix}^{k,n}(x_i, x_i'; \tau_i); f_\theta^{k:K}]) \tag{11}$$

In the backpropagation, we have a process of sampling $\lambda_i$s from the Beta distribution parameterized by $\tau_i$. To enable the backpropagation signals to pass by, we adopt the reparameterization technique of the optimal mass transport (OMT) gradient estimator, which utilizes the implicit differentiation [24, 25], see Appendix C. Finally, the optimal augmentation policy, $\tau_i^*$, of *InfoMixup* for the $i$-th pair of unlabeled data instances, $(x_i, x_i')$, is found as below:

$$\phi^* = \underset{\phi}{\mathrm{argmin}}\, L_\pi([h^k(x_i), h^k(x_i')]), \quad \tau_i^* = \pi_{\phi^*}([h^k(x_i), h^k(x_i')]). \tag{12}$$

### 3.3.2 Acquisition Function by Learned Policy and Model Training

With the optimal policy, $\tau^*$, we construct the acquisition function, $f_{acq}^{LADA}$, which aggregates the acquisition scores of 1) $x_i$, 2) $x_i'$, and 3) their mixed feature maps, $h_{mix}^{k,n}(x_i, x_i'; \tau_i^*)$ as below, with the predicted labels, $\hat{y}$:

$$f_{acq}^{LADA}((x_i, x_i'); f_\theta) = \mathbb{H}[\hat{y}_i|x_i; f_\theta] + \mathbb{H}[\hat{y}_i'|x_i'; f_\theta] + \frac{\gamma}{N}\sum_{n=1}^{N}\mathbb{H}[\hat{y}_i^n|h_{mix}^{k,n}(x_i, x_i'; \tau_i^*); f_\theta^{k:K}]. \tag{13}$$

Assuming that we start the $j^{th}$ iteration of active learning with an already acquired labeled dataset $\mathscr{X}_L^j$, we acquire the set of top-$\frac{b}{2}$ pairs of instances, i.e. $\mathscr{X}_S$, among the subsets, $\mathscr{X}_S' \subset \mathscr{X}_U \times \mathscr{X}_U$ with $|\mathscr{X}_S'| = \frac{b}{2}$, as $\mathscr{X}_S = \mathrm{argmax}_{\mathscr{X}_S' \subset \mathscr{X}_U \times \mathscr{X}_U} \sum_{(x_i, x_i') \in \mathscr{X}_S'} f_{acq}^{LADA}((x_i, x_i'); f_\theta)$. After querying the label of $\mathscr{X}_S$ to *oracle*, we construct a virtual dataset, $\mathscr{X}_M$, using *InfoMixup* with the optimal

---

[1]We denote the forward path from the $i^{th}$ layer to the $j^{th}$ layer of the classifier network as $f_\theta^{i:j}$.

mixing policy, $\tau^*$, as $\mathscr{X}_M = \bigcup_{(x_i, x_i') \in \mathscr{X}_S} \{\lambda_i f_\theta^{0:k}(x_i) + (1-\lambda_i) f_\theta^{0:k}(x_i')\}$, where $\lambda_i \sim \text{Beta}(\tau_i^*, \tau_i^*)$. Here, $\tau^*$ is dynamically inferred by the neural network of $\pi_\phi$ per each pair (see Appendix B.2).

Up to this phase, our training dataset becomes $\mathscr{X}_L^{j+1} = \mathscr{X}_L^j \cup \mathscr{X}_S$ and $\mathscr{X}_M$. Our proposed algorithm, described in Algorithm 1, utilizes $\mathscr{X}_M$ for this active learning iteration only, with various $\lambda_i$s sampled at each training epoch. The classifier network's parameter, $\theta$, is learned via the gradient of the cross-entropy loss as Eq. 14, where $y_i$ denotes the ground-truth label annotated from the *oracle* for the first term; and the mixed label according to the mixing policy for the second term.

$$L_f = \frac{1}{|\mathscr{X}_L^{j+1}|} \sum_{x_i \in \mathscr{X}_L^{j+1}} CE(f_\theta(x_i), y_i) + \frac{1}{|\mathscr{X}_M|} \sum_{x_i \in \mathscr{X}_M} CE(f_\theta^{k:K}(x_i), y_i), \qquad (14)$$

## 4 Experiments

### 4.1 Baselines and Datasets

We specify the instantiated augmentation-acquisition by its subscript, e.g., $\text{LADA}_{\text{EntMix}}$, which adopts *Max Entropy* as data acquisition and *Mixup* as data augmentation. Similarly, we experiment with the various acquisition functions, e.g., *VarMix*, VAALMix, or LL4ALMix, see Appendix D. Also, we experiment with the STN as augmentation policy, e.g., EntSTN, see Appendix E.

We compare our models to 1) Random, 2) BALD [18], 3) Coreset [26], 4) BADGE [27], 5) Max Entropy [16], 6) Variation Ratio [17], 7) VAAL [19] and 8) LL4AL [20]. We also include some data augmented active learning: 1) BGADL [11], 2) CAL [21], 3) *Manifold Mixup* [14] and 4) *AdaMixup* [15]. Here, *Mixup* variants are applied in a pipelined approach. BGADL is also a pipelined approach for the combination, without a learning mechanism in the augmentation from the feedback of acquisition. CAL does not infer the augmentation policy, either. CAL originally aims at semi-supervised learning, so we turn CAL into a supervised setting and we apply *Mixup* after the acquisition by CAL. We also include ablated baselines to see the effect of learning $\tau$, introducing the fixed $\tau$ case, as $\text{LADA}^{\text{fixed}}$.

We conduct experiments on four benchmark datasets: FashionMNIST (Fashion) [28], SVHN [29], CIFAR-10, and CIFAR-100 [30]. Since we assume the scarcity of labeled dataset, we construct a random but balanced initial dataset with 20 instances for Fashion, SVHN, CIFAR-10, and 1,000 instances for CIFAR-100; and we acquire $b$=10 instances for Fashion, SVHN, CIFAR-10, and $b$=100 instances for CIFAR-100 at each iteration, following the prior work [18]. We repeat the acquisition for 100 iterations. To use the same amount of *oracle* queries for all models, we selected top-$\frac{b}{2}$ pairs when adopting *Mixup* as data augmentation in the LADA framework. We set $\gamma$=1 for all experiments in Eq. 5, except for EntSTN with CIFAR-10 as $\gamma$=0.3. We normalize the images with the channel mean and standard deviation over all the datasets. For CIFAR-10 and CIFAR-100 datasets, we apply a standard augmentation, such as random crop and random horizontal flip. We adopt the ResNet-18 [31] as $f_\theta$, and we utilize Adam optimizer [32] with a learning rate of $1e$-03. The policy generator network, $\pi_\phi$ is much smaller network. Appendix A provides details of our experimental settings.

### 4.2 Quantitative Performance Evaluations

Table 1 shows the average test accuracy of five replications, and the accuracy of each replication represents the best accuracy over the acquisition iterations. We separate the performances by the instantiated acquisition functions. The group of baselines does not have any learning mechanism on the acquisition metric. When we examine the general performance gain caused by applying LADA, across datasets, we find the best performers as $\text{LADA}_{\text{EntMix}}$ in Fashion ($\triangle$ 2.52); $\text{LADA}_{\text{VAALMix}}$ in SVHN ($\triangle$ 5.30) and CIFAR-10 ($\triangle$ 4.22); and $\text{LADA}_{\text{LL4ALMix}}$ in CIFAR-100 ($\triangle$ 3.56). In terms of the data augmentation, the *Mixup*-based augmentation outperforms the STN augmentation. In all combinations of baselines and datasets, LADA variations show the best performance in all cases, except the CIFAR-100 with similar performances across CAL, $\text{LADA}_{\text{LL4ALMix}}$ and $\text{LADA}_{\text{VarMix}}$. From the ablation study of $\text{LADA}^{\text{fixed}}$, the learning of the augmentation policy, $\tau$, is meaningful because in 19 out of 20 comparison cases of LADA and $\text{LADA}^{\text{fixed}}$, LADA outperforms $\text{LADA}^{\text{fixed}}$.

Figure 4 shows the test accuracy of the LADA frameworks and the data augmented active learning methods over the acquisition iterations on the CIFAR-10 dataset. The result is averaged over the

Table 1: Comparison of the averaged test accuracy, the run-time of a single iteration of acquisition (Time), and the number of parameters (Param.). The best performance in each category is indicated in boldface. The run-time is calculated as the ratio to the Random acquisition. The number of parameters is only reported for the auxiliary network, and - indicates that no auxiliary network is adopted in the corresponding method. The result is replicated by five-fold.

| | Method | Fashion | SVHN | CIFAR-10 | CIFAR-100 | Time | Param. |
|---|---|---|---|---|---|---|---|
| Baselines | Random | 80.97±0.55 | 71.51±1.41 | 50.15±1.37 | 43.51±0.33 | 1 | - |
| | BALD | 80.79±0.38 | 74.49±3.39 | **54.33±1.23** | 46.29±0.50 | 1.36 | - |
| | Coreset | **83.96±0.55** | **76.89±0.50** | 51.45±0.82 | 43.90±0.76 | 1.54 | - |
| | BADGE | 83.06±0.79 | 75.47±1.87 | 51.83±1.30 | 44.13±0.64 | 1.31 | - |
| | BGADL | 80.47±0.77 | 69.60±1.62 | 45.98±0.73 | 39.33±0.88 | 4.69 | 13M |
| | CAL | 78.10±0.70 | 75.17±2.03 | 53.74±0.89 | **47.38±0.60** | 1.82 | - |
| Entropy-based | Max Entropy | 81.16±1.11 | 72.55±1.21 | 51.45±2.12 | 45.14±0.58 | 1.01 | - |
| | Ent w.ManifoldMixup | 82.03±0.63 | 72.15±1.08 | 51.77±1.76 | 45.96±0.69 | 1.03 | - |
| | Ent w.AdaMixup | 81.29±0.47 | 72.46±1.01 | 51.86±2.32 | 46.23±0.68 | 1.03 | 5K |
| | $LADA_{EntMix}^{fixed}$ | 83.62±0.43 | 74.95±1.30 | 52.77±2.54 | 46.23±0.75 | 1.06 | - |
| | $LADA_{EntMix}$ | **83.68±0.52** | **75.72±1.06** | 53.45±1.67 | 46.92±0.61 | 1.32 | 77K |
| | $LADA_{EntSTN}^{fixed}$ | 81.83±0.26 | 73.03±1.42 | 54.20±1.73 | **45.68±1.73** | 1.02 | 5K |
| | $LADA_{EntSTN}$ | 82.07±0.56 | 73.86±1.09 | **54.95±1.53** | 44.98±1.10 | 1.20 | 5K |
| VarRatio-based | VarRatio | 80.98±0.58 | 73.89±1.08 | 55.88±0.74 | 46.16±0.59 | 1.01 | - |
| | $LADA_{VarMix}^{fixed}$ | 82.84±0.64 | 74.61±0.98 | 56.17±0.73 | 46.54±0.40 | 1.06 | - |
| | $LADA_{VarMix}$ | **83.29±0.27** | **75.24±0.77** | 56.26±1.29 | 47.18±0.97 | 1.33 | 77K |
| | $LADA_{VarSTN}^{fixed}$ | 83.32±0.75 | 74.70±0.75 | 54.22±0.91 | 46.07±0.31 | 1.02 | 5K |
| | $LADA_{VarSTN}$ | **83.35±0.56** | **74.86±1.53** | 55.76±0.53 | **46.42±0.40** | 1.20 | 5K |
| VAAL-based | VAAL | 83.05±0.22 | 72.17±1.85 | 51.05±1.27 | 44.49±0.70 | 3.55 | 301K |
| | $LADA_{VAALMix}^{fixed}$ | 83.77±0.84 | 75.77±0.97 | 53.17±1.13 | 45.98±0.41 | 3.56 | 301K |
| | $LADA_{VAALMix}$ | **84.08±0.41** | **77.47±0.84** | 55.27±1.30 | 46.04±1.09 | 3.60 | 378K |
| | $LADA_{VAALSTN}^{fixed}$ | 83.32±0.77 | 72.86±1.59 | 51.33±0.13 | 44.27±0.26 | 3.56 | 306K |
| | $LADA_{VAALSTN}$ | 83.56±0.53 | 74.53±1.65 | 53.78±2.24 | 45.06±1.29 | 3.57 | 306K |
| LL4AL-based | LL4AL | 83.31±1.34 | 74.14±1.62 | 53.01±2.90 | 43.58±0.42 | 1.55 | 124K |
| | $LADA_{LL4ALMix}^{fixed}$ | 84.59±0.53 | 74.92±1.08 | 55.39±1.49 | 46.88±0.56 | 1.69 | 124K |
| | $LADA_{LL4ALMix}$ | **85.01±0.54** | **76.82±1.64** | 55.73±1.35 | 47.14±0.81 | 1.85 | 201K |
| | $LADA_{LL4ALSTN}^{fixed}$ | **83.69±0.28** | 74.63±1.82 | 53.28±0.67 | 45.01±0.90 | 1.63 | 129K |
| | $LADA_{LL4ALSTN}$ | 83.16±0.22 | **74.74±1.17** | 53.17±0.22 | **45.94±0.61** | 1.68 | 129K |

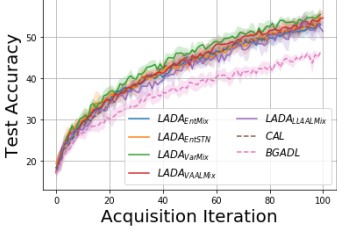
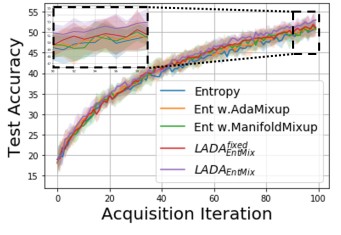
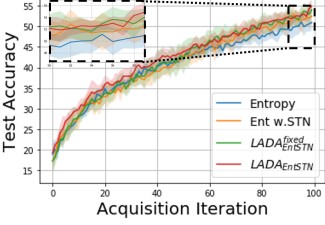

(a) $LADA_{EntMix}$        (b) $LADA_{EntSTN}$

Figure 4: Test accuracy over the acquisition iterations on CIFAR-10 dataset

Figure 5: Ablation study of LADA on CIFAR-10 dataset

five-fold repeated trials, and the shaded area describes the standard deviations. We also provide the figure of the test accuracy on the Fashion, SVHN, and CIFAR-100 datasets in Appendix B.1. Notably, BGADL performs the worst in all datasets, because of the inadequate training of the generative models with the small number of data instances in our active learning setting. The degradation in test accuracies of BGADL becomes apparent as the dataset becomes complex. CAL performs comparably with LADA except the Fashion dataset.

Additionally, we compare the integrated framework, a.k.a. LADA, to the pipelined approaches. In Table 1, *Max Entropy* is the simplest model without an augmentation part. Then, Ent w.*Manifold Mixup* adds the *Manifold Mixup* augmentation, but it does not have a learning process on the mixing policy. Finally, Ent w.*AdaMixup* has a learning process on the mixing policy, but the learning is separated from the acquisition. These pipelined approaches show lower performances than the integration cases of LADA. This ablation study is also shown in Figure 5; *Mixup*-based augmentations in Figure 5a and STN-based augmentations in Figure 5b, respectively. The figures confirm the effects

of 1) considering the gain of augmentation in the acquisition functions, as well as 2) learning the augmentation policy with feedback from the acquisition.

Next, since LADA trains the augmentation policy, we also compare LADA with AutoAugment [33]. To show the effectiveness of the look-ahead concept, we trained AutoAugment and then applied it in 1) pipe-lined approach (a.k.a. Ent w.AA) and 2) LADA approach (a.k.a. $\text{LADA}^{\text{fixed}}_{\text{EntAA}}$). As shown in Table 2, the $\text{LADA}^{\text{fixed}}$ shows better performance than the pipelined approach case, indicating that looking ahead the informativeness of the virtual instances yields good performance. However, it should be noted that training AutoAugment requires the labeled dataset, which is not available in the active learning setting. Therefore, the learned AutoAugment could have been cheating under the assumption of joint acquisition and augmentation. Second, the pre-trained AutoAugment does not select the augmentation to maximize the acquisition score of the unlabeled instances, so the acquisition with AutoAugment would not be optimal considering the missing contribution of labeling hard instances. Some hard instances may become very informative after the acquisition because of the augmented virtual instances, but the learned AutoAugment cannot anticipate this opportunity of information gain because it is pre-trained and static.

Table 2: Test accuracy of pipe-lined method and LADA with learned AutoAugment

| Method | Fashion | SVHN | CIFAR-10 | CIFAR-100 |
|---|---|---|---|---|
| Ent w.AA | $84.77 \pm 1.12$ | $75.40 \pm 1.28$ | $53.44 \pm 1.94$ | $46.78 \pm 1.23$ |
| $\text{LADA}^{\text{fixed}}_{\text{EntAA}}$ | $85.09 \pm 0.19$ | $77.44 \pm 2.97$ | $55.22 \pm 0.86$ | $48.31 \pm 1.70$ |

Also, we extend LADA by applying it to semi-supervised learning, since semi-supervised learning algorithms also rely on the augmentation. For this experiment, we adopt the $\Pi$-model [34] for semi-supervised model. As shown in Table 3, the combination of $\Pi$-model with LADA shows better performance than the combination with Entropy acquisition.

Table 3: Test accuracy of semi-supervised learning with LADA on CIFAR-10 dataset

| Methods | # of labeled samples | | | | |
|---|---|---|---|---|---|
| | 250 | 500 | 1000 | 2000 | 4000 |
| $\Pi$-model + Entropy | 45.47 | 56.40 | 66.09 | 75.46 | 81.61 |
| $\Pi$-model + $\text{LADA}^{\text{fixed}}_{\text{EntMix}}$ | 45.47 | 59.51 | 68.30 | 77.77 | 82.86 |
| $\Pi$-model + $\text{LADA}_{\text{EntMix}}$ | 45.47 | 58.94 | 68.92 | 78.54 | 82.97 |

### 4.3 Qualitative Analysis on Acquired Data Instances

Besides the quantitative comparison, we provide reasoning on the behavior of LADA. Therefore, we select $\text{LADA}_{\text{EntMix}}$ to contrast to the pipelined approach. We investigate on 1) selecting the informative data instances by acquisition, 2) validating the optimal $\tau^*$ in the augmentation learned from the policy generator network $\pi_\phi$, and 3) examining the coverage of the explored space.

To check the informativeness of data instances, Figure 6 shows the different process of acquiring instances between *Max Entropy* and $\text{LADA}_{\text{EntMix}}$. *Max Entropy* selects a data instance with the highest predictive entropy value. Compared to *Max Entropy*, $\text{LADA}_{\text{EntMix}}$ selects a pair of two data instances with the highest value of the aggregated predictive entropy, i.e., the summation of the predictive entropy from two data instances and one *InfoMixup* instance, as Eq. 13. By mixing two unlabeled data instances with the optimal mixing policy $\tau^*$, the virtual data instance, generated along the interpolated space, results in a higher entropy value than the selected instance by *Max Entropy*.

To confirm the validity of the optimal $\tau^*$, we compare three cases of 1) the inferred $\tau$ ($\text{LADA}_{\text{EntMix}}$); 2) the fixed $\tau$ ($\text{LADA}^{\text{fixed}}_{\text{EntMix}}$); and 3) the pipelined model's $\tau$ (Ent w.*Manifold Mixup*). Figure 7a shows the entropy of the virtual data instances over the acquisition process. The optimal $\tau^*$ inferred in $\text{LADA}_{\text{EntMix}}$ produces the highest entropy over the acquisition process. The differentiation becomes significant after some acquisition iterations, which comes from the requirement of training the classifier. Figure 7b shows the entropy distribution of virtual instances, with the median value of each interval as $x$-axis. This also shows that the optimal $\tau^*$ has the highest density beyond the interval of the median 2.2.

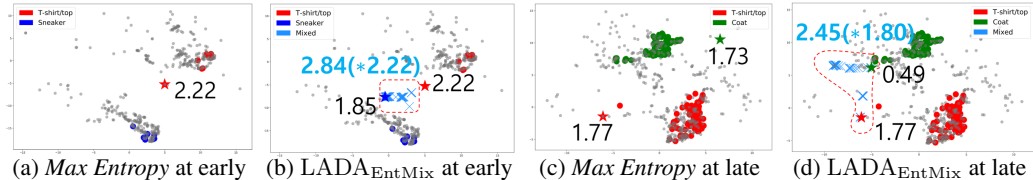

(a) *Max Entropy* at early  (b) LADA$_{\text{EntMix}}$ at early  (c) *Max Entropy* at late  (d) LADA$_{\text{EntMix}}$ at late

Figure 6: tSNE [35] plot of acquired instance ($\star$) and augmented instance ($\times$), with entropy values. The numbers written in *black* indicate the predictive entropy of unlabeled data instances that were selected from the unlabeled pool. The numbers written in *blue* indicate the maximum (*average) value of predictive entropy of the virtual data instances that were generated from *InfoMixup*. The acquisition iterations of early and late are 7 and 76, respectively.

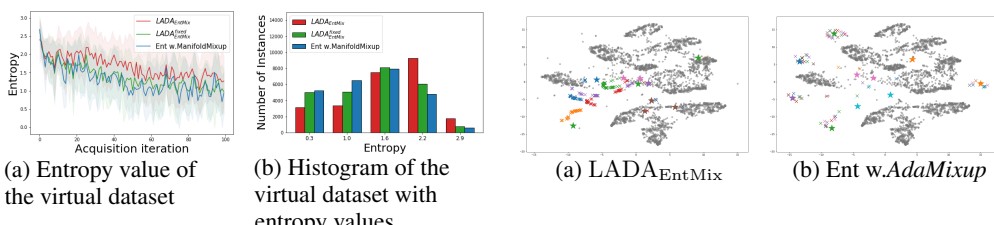

(a) Entropy value of the virtual dataset

(b) Histogram of the virtual dataset with entropy values

(a) LADA$_{\text{EntMix}}$          (b) Ent w.*AdaMixup*

Figure 7: Entropy values of the virtual data generated from LADA$_{\text{EntMix}}$, LADA$_{\text{EntMix}}^{\text{fixed}}$, and Ent w.*Manifold Mixup*

Figure 8: tSNE plot of acquired data instances ($\star$) and generated virtual data instances ($\times$). The labels are categorized by colors.

To examine the coverage of the explored latent space, Figure 8 illustrates the latent space of the acquired data instances and the augmented data instances. Ent w.*AdaMixup* learns the policy $\tau$ to avoid the manifold intrusion, so its learned $\tau$ limits a sample of $\lambda$ to be placed near either one of the paired instances. Therefore, Ent w.*AdaMixup* ends up exploring the space near the acquired instances. In contrast, the generated virtual instances by LADA$_{\text{EntMix}}$ show further exploration, because the optimal $\tau^*$ is guided by the entropy maximization and seeks along with the space that has not been explored by the model yet. The latent space makes the linear interpolation of LADA$_{\text{EntMix}}$ to be curved by the manifold, but it keeps the interpolation line of the curved manifold.

## 5    Conclusions

In the real world, gathering a large-scale labeled dataset is difficult, and learning a deep neural network requires effective utilization of the limited resources. This limitation motivates the integration of data augmentation and active learning. In this paper, we propose a generalized framework for such integration, named as LADA, which adaptively selects the informative data instances by looking ahead the acquisition score of both 1) unlabeled data instances and 2) virtual data instances to be generated by data augmentation, in advance of the acquisition process. To enhance the effect of the data augmentation, LADA learns the augmentation policy to maximize the acquisition score of the virtual instance, as well. Through quantitative and qualitative analysis with various instantiations, LADA is confirmed to select and augment informative data instances.

## 6    Limitations and Ethical Discussion

The proposed work is limited to the image classification task, so other tasks, e.g., object detection and semantic segmentation, need to be studied in the future. LADA can be applied to these tasks with simple extension in augmentations, and such extension will be the main topic of the study. However, LADA still maintains its structure by differentiating the implemented augmentation policies by tasks. On the societal impact, privacy issue is concerned when selecting and labeling dataset. Also, we need to check the robustness of LADA to prevent the failure modes or sensitivities to architectural choices.

## 7    Acknowledgement

This work was supported by the Technology development Program (S3125937) funded by the Ministry of SMEs and Startups (MSS, Korea).

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
