# Supplementary Materials of
# LADA: Look-Ahead Data Acquisition via Augmentation for Active Learning

**Yoon-Yeong Kim**[1]    **Kyungwoo Song**[2]    **Joonho Jang**[1]    **Il-Chul Moon**[1,3]
[1]Korea Advanced Institute of Science and Technology (KAIST)    [2]University of Seoul    [3]Summary.AI
yoonyeong.kim@kaist.ac.kr kyungwoo.song@uos.ac.kr
adkto8093@kaist.ac.kr icmoon@kaist.ac.kr

## A   Experimental Settings

### A.1   URL of Code Implementation

Our code and data are available at `https://github.com/YoonyeongKim/LADA`.

### A.2   Active Learning Scenario

Table 1 shows the details of the active learning scenario and the hyper-parameters that are used in LADA or baseline models. It should be noted that when adopting Variational Adversarial Active Learning (VAAL) [1] and Learning Loss for Active Learning (LL4AL) [2] as the acquisition function in LADA, we use *Mixup* att the input level with pixels, which will be discussed in Section D. The initial budget is a random but balanced labeled set for the initial training of the classifier network, $f_\theta$. At each acquisition iteration, we randomly sub-sample 2,000 data instances from the unlabeled pool to calculate the acquisition score, following the prior work [3]. After calculating the acquisition score of the unlabeled data instances, we select top-$b$ instances according to the acquisition score, where $b$ indicates the budget.

Table 1: Details of active learning scenario in each dataset

| Dataset | Initial Budget (Data num. of each class) | Budget $b$ | Acquisition Iterations | $\tau$ in fixed *Manifold Mixup* (*Mixup* of input level) | $N$ of sampling $\lambda$ in LADA with *Manifold Mixup* |
|---|---|---|---|---|---|
| Fashion | 20 (2) | 10 | 100 | 2.0 (0.2) | 10 |
| SVHN | 20 (2) | 10 | 100 | 2.0 (0.2) | 1 |
| CIFAR-10 | 20 (2) | 10 | 100 | 2.0 (0.2) | 10 |
| CIFAR-100 | 1000 (10) | 100 | 100 | 2.0 (0.2) | 5 |

When applying LADA framework with *Mixup* as the augmentation, we randomly paired the sub-sampled 2,000 data instances, resulting in the construction of 1,000 pairs. Then, we learned the mixing policy, $\tau_i$, for the $i$-th pair. After learning the policy, we select top-$\frac{b}{2}$ pairs of data instances to equally utilize the budget of querying the *oracle*.

### A.3   Network Structure and Training Details

The classifier network, $f_\theta$, adopts 18-layer residual network (Resnet-18) [4]. We utilize the Adam optimizer [5] with a learning rate of $1e$-03. After the acquisition, the classifier network is trained for 50 epochs, following the prior work [3]. The batch size for training the classifier network is 10 for Fashion, SVHN, CIFAR-10; and 100 for CIFAR-100.

35th Conference on Neural Information Processing Systems (NeurIPS 2021), Sydney, Australia.

The policy generator network, $\pi_\phi$, consists of two convolutional layers with the ReLU activation followed by the max-pooling layer, and three fully connected layers. To construct the vicinal space of the real data instances carefully, we use the sigmoid function as the activation function of the last fully-connected layer of the policy generator network. Using the sigmoid function also makes it possible to constraint the value of $\tau$ to be non-negative. Table 2 and Table 3 show the details of the network structure in each dataset. It should be noted that the policy generator network is a simple neural network compared to the classifier network. When randomly selecting the $k$-th layer to perform *Manifold Mixup*, we choose from the output layer of each block in the classifier network, following the prior work [6]. The output shape of each block in Resnet-18 is the same among each block, so the shape of the feature maps that are put into the policy generator network, $\pi_\phi$, is maintained the same. The training procedure of $\pi_\phi$ utilizes the Adam optimizer with a learning rate of $5e$-05, and we train $\pi_\phi$ for 10 epochs for a given batch. The batch size of training the policy generator network is 100 for all dataset.

Table 2: Structure of the policy generator network, $\pi_\phi$, in Fashion dataset

| Layer | Type | Input | Kernel Num. | Kernel Size | Stride | Padding | Activation | Output |
|---|---|---|---|---|---|---|---|---|
| 1 | Convolution1 | 128×28×28 | 6 | 5×5 | 1 | 2 | ReLU | 6×28×28 |
| 2 | Max-Pooling | 6×28×28 | – | 2×2 | 2 | – | – | 6×14×14 |
| 3 | Convolution2 | 6×14×14 | 16 | 5×5 | 1 | – | ReLU | 16×10×10 |
| 4 | Max-Pooling | 16×10×10 | – | 2×2 | 2 | – | – | 16×5×5 |
| 5 | Flatten | 16×5×5 | – | – | – | – | – | 400 |
| 6 | FC1 | 400 | – | – | – | – | Tanh | 120 |
| 7 | FC2 | 120 | – | – | – | – | Tanh | 60 |
| 8 | FC3 | 60 | – | – | – | – | Sigmoid | 1 |

Table 3: Structure of the policy generator network, $\pi_\phi$, in SVHN, CIFAR-10 and CIFAR-100 dataset

| Layer | Type | Input | Kernel Num. | Kernel Size | Stride | Padding | Activation | Output |
|---|---|---|---|---|---|---|---|---|
| 1 | Convolution1 | 128×32×32 | 6 | 5×5 | 1 | 2 | ReLU | 6×32×32 |
| 2 | Max-Pooling | 6×32×32 | – | 2×2 | 2 | – | – | 6×16×16 |
| 3 | Convolution2 | 6×16×16 | 16 | 5×5 | 1 | – | ReLU | 16×12×12 |
| 4 | Max-Pooling | 16×12×12 | – | 2×2 | 2 | – | – | 16×6×6 |
| 5 | Flatten | 16×6×6 | – | – | – | – | – | 576 |
| 6 | FC1 | 576 | – | – | – | – | Tanh | 120 |
| 7 | FC2 | 120 | – | – | – | – | Tanh | 60 |
| 8 | FC3 | 60 | – | – | – | – | Sigmoid | 1 |

## A.4   Complexity Analysis

Table 4 reports the computational complexity of *Max Entropy* and $\text{LADA}_{\text{EntMix}}$, in terms of scoring, sorting, and training. In Table 4, $U$ is the size of the unlabeled dataset, $L$ is the size of the labeled dataset, $R$ is the complexity of feed-forwarding through the classifier network and $G$ is the complexity of feed-forwarding through the policy generator network. We utilized Titan X GPUs to implement our framework.

Table 4: Computational complexity of *Max Entropy* and $\text{LADA}_{\text{EntMix}}$, in big-O notation. Scoring denotes the calculation of the acquisition score for the unlabeled dataset. Sorting denotes the quick-sort algorithm to sort the unlabeled data instances by the acquisition scores. Training denotes the learning of the classifier network with the labeled dataset, which is selected by the acquisition score; and annotated by the *oracle*.

| | Scoring | Sorting | Training |
|---|---|---|---|
| *Max Entropy* | $O(UR)$ | $O(U \log U)$ | $O(LR)$ |
| $\text{LADA}_{\text{EntMix}}$ | $O(UR + UG)$ | $O(U \log U)$ | $O(LR)$ |

# B Experimental Results

## B.1 Test accuracy on Fashion, SVHN and CIFAR-100 Dataset

Figure 1 compares the LADA framework with other data augmented active learnings, on Fashion, SVHN, and CIFAR-100 datasets.

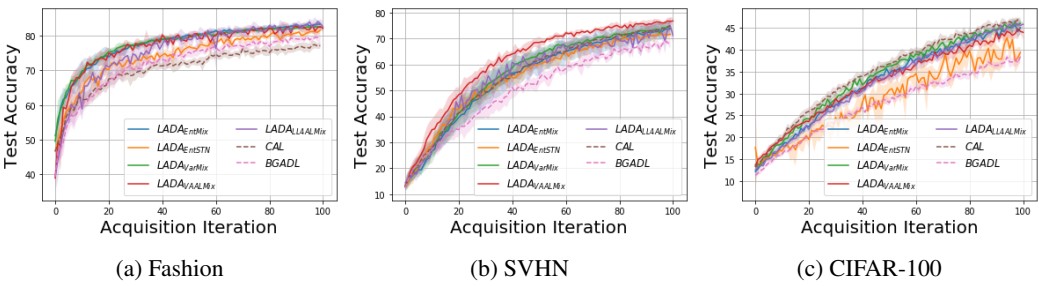

(a) Fashion      (b) SVHN      (c) CIFAR-100

Figure 1: Test accuracy over the acquisition iterations on Fashion, SVHN and CIFAR-100 dataset

## B.2 Learning of the Policy Generator Network

Figure 2 shows the value of $\tau^*$ that is dynamically inferred over the acquisition iterations. In the relatively simple dataset such as Fashion, the inferred $\tau^*$ is close to 1.0, which results in a less-sharp shape of Beta distribution. Hence, LADA explores broad space between two instances of the pair. Meanwhile, in the complex dataset such as SVHN, CIFAR-10 and CIFAR-100, $\tau^*$ is inferred as a relatively small value at the initial iterations, which results in a sharp shape of Beta distribution. Hence, at the initial iterations, LADA generates virtual instances in the vicinity of each data instance of the pair. However, after some iterations of training the classifier network, the inferred $\tau^*$ increases and LADA explores more broad space between the two instances of the pair to find more informative data instances, i.e. data instances with high predictive entropy value.

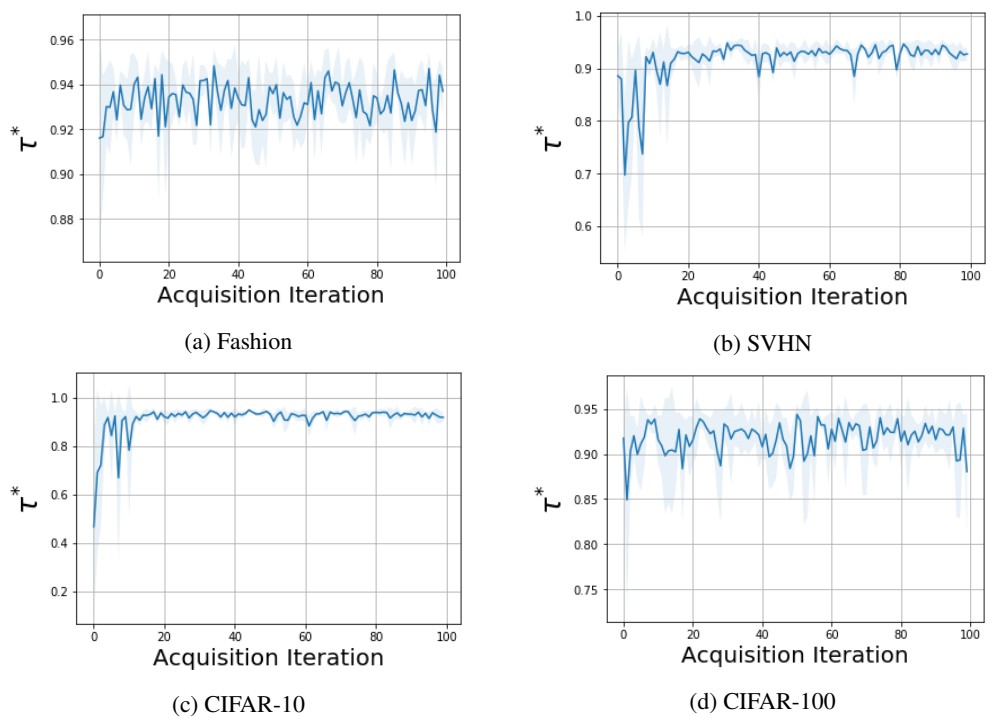

(a) Fashion      (b) SVHN

(c) CIFAR-10      (d) CIFAR-100

Figure 2: Optimal $\tau^*$ over the acquisition iterations, inferred by the policy generator network, $\pi_\phi$

## B.3 Test accuracy with Larger Number of Data

To show the robustness of the LADA framework, we also experiment with the larger number of instances, following the setting of the prior work of [2]. For CIFAR-10 dataset, we construct the initial labeled dataset with 1,000 instances. The budget size, $b$, per acquisition is 1,000 and we repeat the acquisition until we collect 10,000 labeled dataset. For CIFAR-100 dataset, the initial labeled dataset consists of 5,000 instances. The budget size, $b$, per acquisition is 2,500 and the acquisition iteration is repeated until the size of the labeled dataset becomes 20,000. We train the classifier network for 50 epochs after each acquisition, with the Adam optimizer and the learning rate of $1e$-03. We experiment by adopting VAAL and LL4AL as $f_{acq}$, and *Mixup* as $f_{aug}$, for the LADA framework. Figure 3 shows that the performance gain by the LADA framework is apparent when the number of labeled instances are small, which corresponds to the earlier iterations of the acquisition. Table 5 reports the averaged test accuracy of each replication for the baselines and the LADA frameworks, and the accuracy of each replication represents the best accuracy over the acquisition iterations.

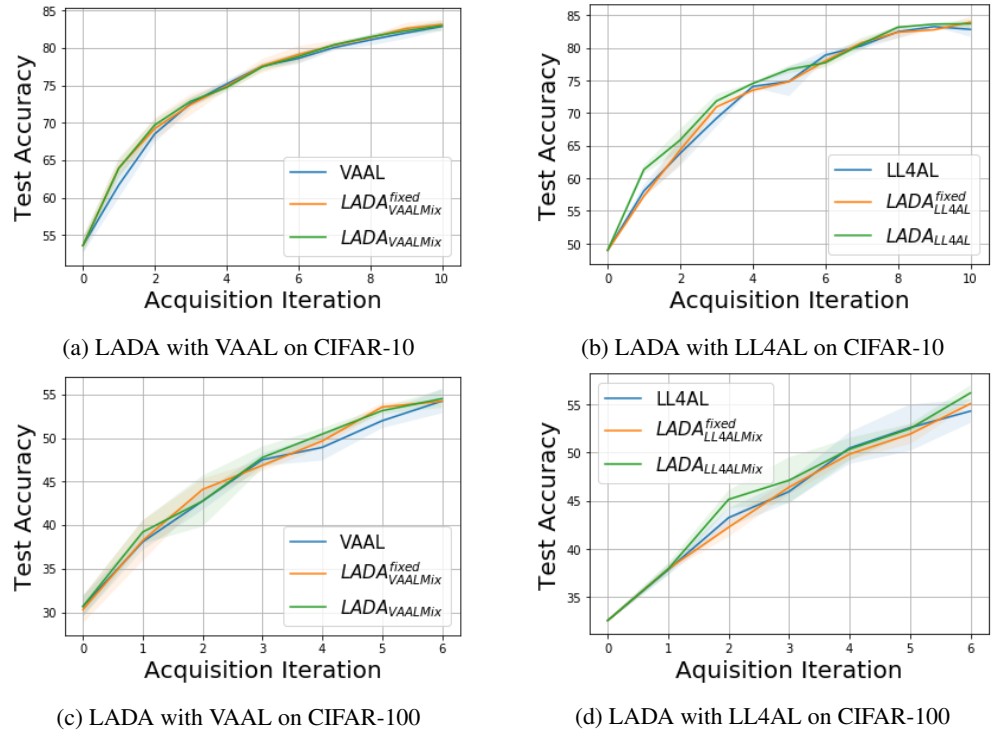

(a) LADA with VAAL on CIFAR-10

(b) LADA with LL4AL on CIFAR-10

(c) LADA with VAAL on CIFAR-100

(d) LADA with LL4AL on CIFAR-100

Figure 3: Test accuracy over the acquisition iterations on CIFAR-10 and CIFAR-100 datasets, with the larger number of data

Table 5: Comparison of the averaged test accuracy on CIFAR-10 and CIFAR-100 dataset with the larger number of data

| Method | | CIFAR-10 | CIFAR-100 |
|---|---|---|---|
| | VAAL | 82.90±0.52 | 53.85±1.40 |
| VAAL-bsed | $LADA^{fixed}_{VAALMix}$ | 83.29±0.36 | 54.39±1.36 |
| | $LADA_{VAALMix}$ | **83.32±0.31** | **54.64±0.73** |
| | LL4AL | 83.58±0.36 | 54.60±1.24 |
| LL4AL-based | $LADA^{fixed}_{LL4ALMix}$ | 83.94±0.88 | 55.06±0.13 |
| | $LADA_{LL4ALMix}$ | **84.52±0.87** | **56.17±0.78** |

## C   Optimal Mass Transport Gradient Estimator

In the backpropagation for training the policy generator network $\pi_\phi$, we have a process of sampling $\lambda$ from the Beta distribution parameterized by $\tau$. To enable the backpropagation signals to pass by, we need the pathwise gradient estimator, or the reparameterization tricks.

Suppose we designate the Beta distribution parameterized by $\tau$ from which we sample $\lambda$, as $q_\phi(\lambda)$, and designate the objective function originated from $\lambda$ as $f_\phi(\lambda)$, or $L$. Here, $\phi$ refers to the parameters of the policy generator network $\pi_\phi$. What we have to compute is the derivative of $L$ w.r.t. the parameter $\phi$ as below.

$$\nabla_\phi L = \nabla_\phi E_{q_\phi(\lambda)}[f_\phi(\lambda)] \tag{1}$$

In the reparameterization trick, the continuous random variable whose probability density $q_\phi(\lambda)$ is reparameterized such that we can rewrite expectations as below.

$$E_{q_\phi(\lambda)}[f_\phi(\lambda)] \rightarrow E_{q_0(\epsilon)}[f_\phi(g(\epsilon; \phi))] \tag{2}$$

Since the expectation w.r.t. $q_0(\epsilon)$ does not depend on $\phi$, gradients w.r.t. $\phi$ can be computed. However, the reparameterization trick is non-trivial to apply to the Beta distribution, since the required shape transformation $g(\epsilon; \phi)$ for the Beta distribution does not have special functions.

To deal with this, we use the optimal mass transport (OMT) gradient estimator, which utilizes the implicit differentiation. By making use of implicit differentiation, the gradient of Eq.(1) is computed as below [7, 8].

$$\nabla_\phi L = E_{q_\phi(\lambda)}\left[\frac{df_\phi(\lambda)}{d\lambda}\frac{d\lambda}{d\phi} + \frac{\partial f_\phi(\lambda)}{\partial\phi}\right] \tag{3}$$

$$\frac{d\lambda}{d\phi} = -\frac{\frac{\partial F_\phi}{\partial\phi}(\lambda)}{q_\phi(\lambda)} \tag{4}$$

In Eq.(3) $\sim$(4), the key ingredient needed to compute the gradient is computing the derivative of the CDF, or the $\frac{\partial F_\phi}{\partial\phi}(\lambda)$. For Beta distribution in the policy generator network $\pi_\phi$, the CDF of the distribution is given by $F_{\tau,\tau}(\lambda) = \frac{B(\lambda;\tau,\tau)}{B(\tau,\tau)}$ where $B(\lambda;\tau,\tau)$ is the incomplete beta function and $B(\tau,\tau)$ is the beta function. Then, by using a Taylor series expansion, we can compute $B(\lambda;\tau,\tau)$ in powers of $\lambda$ as below.

$$B(\lambda;\tau,\tau) = \lambda^\tau\left(\frac{1}{\tau} + \frac{1-\tau}{1+\tau}\lambda + \frac{1 - \frac{3\tau}{2} + \frac{\tau^2}{2}}{2+\tau}\lambda^2 + \cdots\right) \tag{5}$$

This allows us to compute the derivatives of the beta function w.r.t. $\tau$ and further the derivative of Eq.(3).

Finally, we use **rsample** function in PyTorch for the implementation of the OMT. Representing the policy generator network $\pi_\phi$ as MLP, the implementation code of sampling $\lambda$ from the Beta distribution parameterized by $\tau$ is as below.

```
tau = MLP(x1,x2)
f = distribution.beta.Beta(tau, tau)
lambda = f.rsample()
```

# D LADA with Various Acquisition

Since the LADA framework is proposed as a generalized framework that can adopt various types of acquisition and augmentation functions, we apply various types of acquisition and augmentations.

In VAAL, the acquisition score of each data instance is calculated independently by the current classifier network, $f_\theta$. Also, the discriminator, $\varphi^{VAAL}$, is trained with the VAE's latent variable $z$ as an input, and VAE is trained with raw-level data instances, $x$. For LL4AL, the acquisition score is calculated via the loss prediction module, $f_{LPM}$, which receives depth-wise features from the classifier network, $f_\theta$, as input. Hence, we use the input feature *Mixup* as the data augmentation for LADA with these two acquisition functions.

Eq. 6 and Eq. 7 are the learning objective of data augmentation policy, $\tau$, of LADA with VAAL, i.e., $\text{LADA}_{\text{VAALMix}}$, and LADA with LL4AL, i.e., $\text{LADA}_{\text{LL4ALMix}}$, respectively, where $\lambda_i \sim \text{Beta}(\tau_i, \tau_i)$. Eq. 6 and Eq. 7 are analogous to the objective of LADA with *Max Entropy*, which we mainly described in Section 3.3.1.

$$
\begin{aligned}
\tau_i^* &= \underset{\tau}{\operatorname{argmax}}\, f_{acq}^{VAAL}(\lambda_i x_i + (1 - \lambda_i)x_i'; \varphi^{VAAL}) \\
&= \underset{\tau}{\operatorname{argmax}}\, \mathrm{P}(\lambda_i x_i + (1 - \lambda_i)x_i' \in \mathscr{X}_U; \varphi^{VAAL})
\end{aligned}
\tag{6}
$$

$$
\begin{aligned}
\tau_i^* &= \underset{\tau}{\operatorname{argmax}}\, f_{acq}^{LL4AL}(\lambda_i x_i + (1 - \lambda_i)x_i'; f_{LPM}) \\
&= \underset{\tau}{\operatorname{argmax}}\, f_{LPM}(f_\theta^k(\lambda_i x_i + (1 - \lambda_i)x_i')|k \in K)
\end{aligned}
\tag{7}
$$

Figure 4 illustrates the inferring process of the augmentation policy, $\tau$, through the policy generator network, $\pi_\phi$: Figure 4a for $\text{LADA}_{\text{VAALMix}}$ and Figure 4b for $\text{LADA}_{\text{LL4ALMix}}$, respectively. We introduce a simplifying notation, $x_{mix}$, to represent $\lambda_i x_i + (1 - \lambda_i)x_i'$. Also, $\hat{L}(x_{mix})$ denotes the predictive loss of the mixed instance, $x_{mix}$, which is the output of the loss prediction module, $f_{LPM}$, of LL4AL.

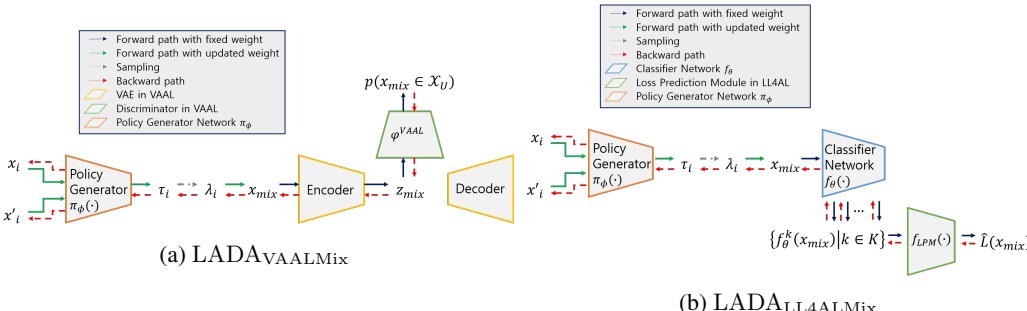

(a) $\text{LADA}_{\text{VAALMix}}$

(b) $\text{LADA}_{\text{LL4ALMix}}$

Figure 4: Process of inferring $\tau$ through $\pi_\phi$ in LADA with various acquisition functions

# E   LADA with STN Augmentation

As described in Section 3.2.1, STN consists of 1) a loclization network, $f_\tau$, i.e., a neural network parameterized by $\tau$, 2) a grid generaotor function, $f_T$, and 3) a sampler function, $f_S$. The augmentation policy of STN that is trained in LADA framework is the parameters of the localization network, $f_\tau$. The transformation of the data instances, $x$, into $\tilde{x}$, is as below:

$$\nu = f_\tau(x), \tag{8}$$
$$g = f_T(G; \nu), \tag{9}$$
$$\tilde{x} = f_{aug}^{STN}(x; \tau) = f_S(x, g) = f_S(x, f_T(G; \nu)) \tag{10}$$
$$= f_S(x, f_T(G; f_\tau(x))). \tag{11}$$

The structure of STN is described in Figure 5.

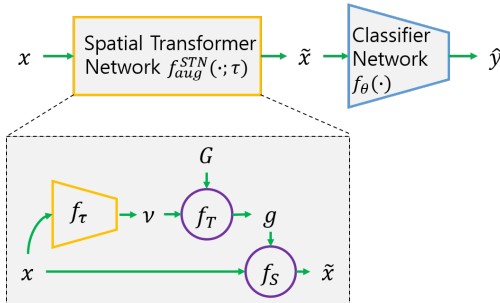

Figure 5: Process of STN augmentation, inserted in the classifier network, $f_\theta$

Algorithm 1 shows the detailed process of LADA framework with *Max Entropy* as acquisition function, $f_{acq}$, and STN as augmentation policy, $f_{aug}$.

---

**Algorithm 1** LADA with *Max Entropy* and STN

---

**Input:** Labeled dataset $\mathscr{X}_L^0$, Classifier $f_\theta$, STN $f_{aug}^{STN}(\cdot; \tau)$
1: **for** $j = 0, 1, 2, \dots$ **do**                                              ▷ active learning
2:     Save the current augmentation policy of the STN, $\tau$
3:     Randomly sample $\mathscr{X}_U^{pool} \subset \mathscr{X}_U$
4:     $\tau^* = \mathrm{argmax}_\tau \frac{1}{|\mathscr{X}_U|} \sum x_U \in \mathscr{X}_U \mathbb{H}[\hat{y} | f_{aug}^{STN}(x_U; \tau); f_\theta]$
5:     Select $\mathscr{X}_S = \mathrm{argmax}_{\mathscr{X}_S' \subset \mathscr{X}_U} \sum_{x \in \mathscr{X}_S'} \left( \mathbb{H}[\hat{y}|x; f_\theta] + \mathbb{H}[\hat{y}|f_{aug}^{STN}(x; \tau^*); f_\theta] \right)$,
           with $|\mathscr{X}_S'| = b$
6:     Query the selected dataset, $\mathscr{X}_S$
7:     Update the labeled datset, $\mathscr{X}_L^{j+1} = \mathscr{X}_L^j \cup \mathscr{X}_S$
8:     Augment the selected dataset, $\tilde{\mathscr{X}}_S = f_{aug}^{STN}(\mathscr{X}_S; \tau^*)$
9:     Load the saved parameters of the STN, $\tau$
10:    **for** $t = 0, 1, 2, \dots$ **do**                              ▷ training $f_\theta$ and $f_{aug}^{STN}(\cdot; \tau)$
11:        Update $\theta$ and $\tau$ with cross-entropy loss of $\mathscr{X}_L^{j+1}$, in end-to-end fashion
12:        Update $\theta$ with cross-entropy loss of $\tilde{\mathscr{X}}_S$
13:    **end for**
14: **end for**

---

# F  tSNE Plot of Data Instances

This section shows the different behavior between *Max Entropy* and LADA during the active learning iteration $i$ for various dataset. In each figure, the numbers written in *black* represent the predictive entropy of the unlabeled data instance ($\star$) that were selected from the unlabeled pool. The numbers written in *red* represent the maximum (*average) entropy of the virtual data instance ($\times$) that were generated from *InfoMixup*.

## F.1  Fashion

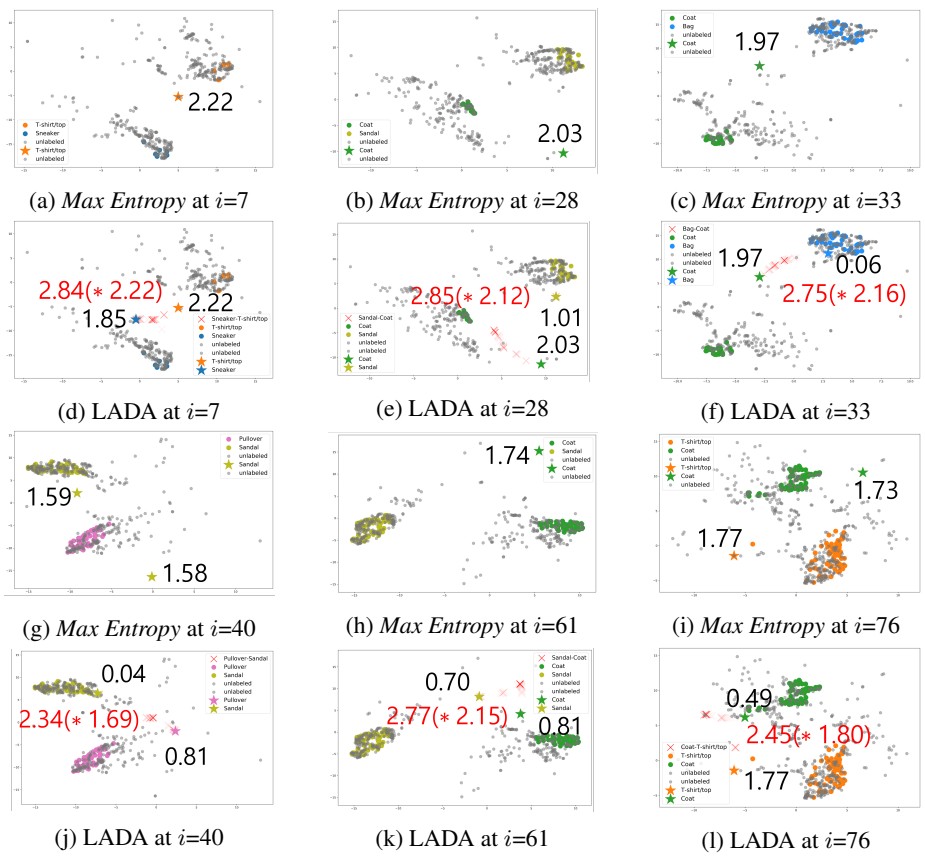

Figure 6: tSNE plot of acquired instances ($\star$) and augmented instances ($\times$) in Fashion dataset

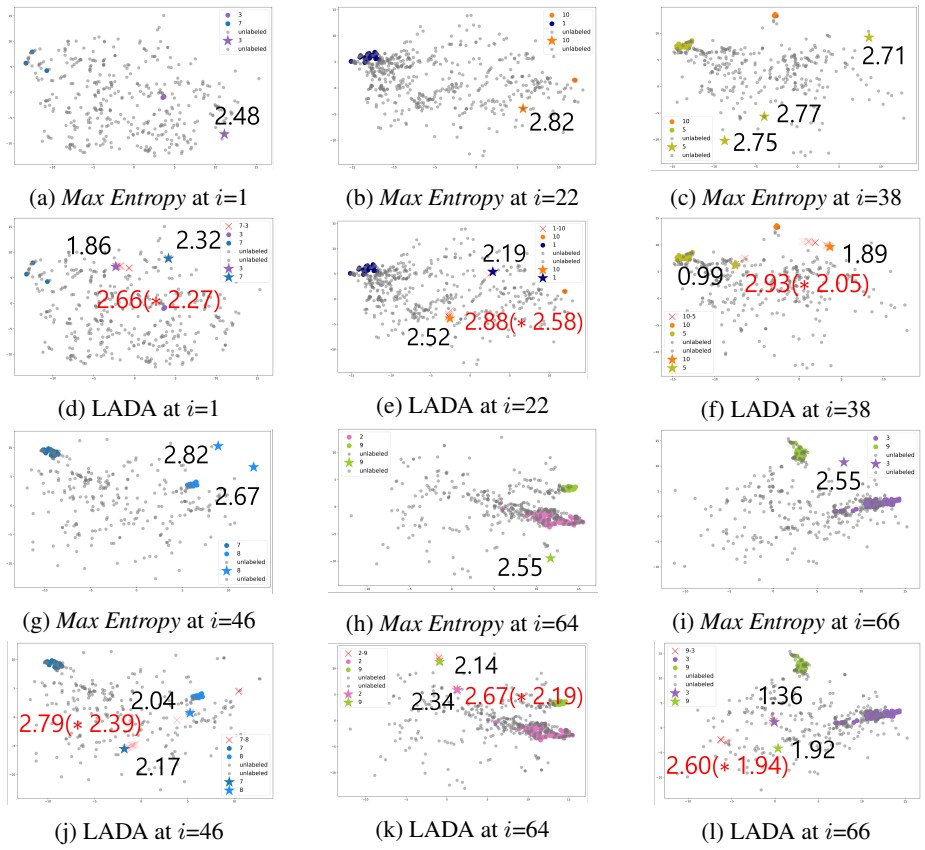

Figure 7: tSNE plot of acquired instances ($\star$) and augmented instances ($\times$) in SVHN dataset

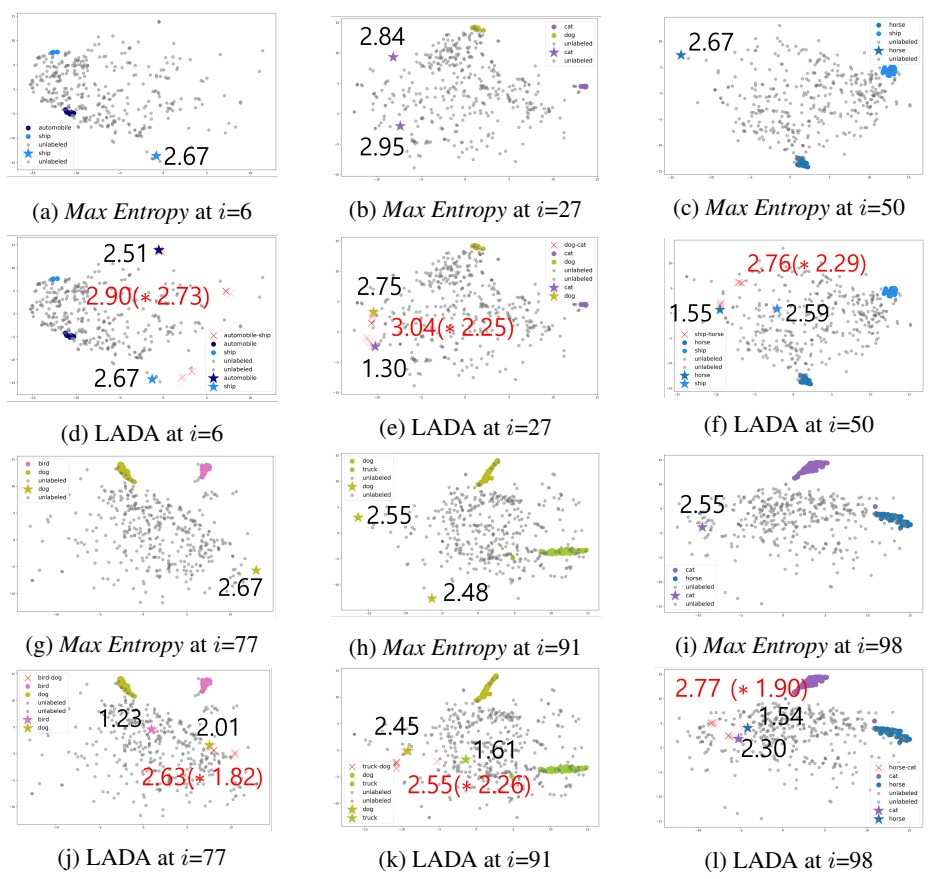

Figure 8: tSNE plot of acquired instances ($\star$) and augmented instances ($\times$) in CIFAR-10 dataset