# OpenReview forum: "LADA: Look-Ahead Data Acquisition via Augmentation for Deep Active Learning"
_NeurIPS.cc/2021/Conference — NeurIPS 2021 Poster_

### Official Review · Reviewer_pwAg · 2021-07-12

**Rating:** 6
**Confidence:** 4

**Summary:**

The paper proposes to combine data augmentation with active learning to improve the performance of Active Learning approaches. Instead of a pipelined approach, where the augmentation is applied on top of the data selected by the acquisition function, the paper aims to involve data augmentation in the data acquisition step too. Data augmentation policy is optimized to increase the acquisition function score, and the resulting augmented data is then used in the search process.



**Limitations And Societal Impact:**

Yes, the authors have addressed it.

**Main Review:**

Pros -
The paper does a nice job in motivating the problem, and also the introduction of approach is clear and concise.

Cons -

1. I don't think the paper considers good baselines. If the paper is learning augmentations, then the $LADA^{fixed}$ baseline should be done over stronger augmentations, such as Auto-augment [1] where the augmentation policies have been learnt from before.

2. The baselines for active learning are also not strong. The authors should include results with Core-Set baseline[2] which performs better than VAAL, entropy based and VarRatio-based approaches.

3. The experimental setup should also include results for more iterations of active learning, probably till 10k samples selected for labelling for CIFAR-10 as done in [2] and [3].

4. Section 3.3.1 which is one of the important sections of the paper is not explained properly. Its hard to understand how manifold mixup augmentation is used for LADA.

References -
[1] AutoAugment: Learning Augmentation Policies from Data. Cubuk et al.
[2] Active Learning for Convolutional Neural Networks: A Core-Set Approach. Sener et al.
[3] Learning Loss for Active Learning. Yoo et al.

**Time Spent Reviewing:**

3

---

> ### Author Response · Authors · 2021-08-04
> **Response to Reviewer pwAg**
>
> Thanks so much for your constructive reviews. Please see our response below.
>
>
> 1. Answer to Cons 1
>
> Following the reviewer’s suggestion, we conducted the experiments using the learnt AutoAugment, with the CIFAR-10 dataset and Entropy-based acquisition, during the rebuttal period. To show the effectiveness of the look-ahead concept, the experiment uses the learnt AutoAugment as 1) pipe-lined approach (a.k.a. Ent w. AA) and 2) data augmentation component of LADA$^{fixed}$ (a.k.a. LADA$^{fixed}_{EntAA}$). The LADA$^{fixed}$ case shows better performance than the pipelined approach case, indicating that looking ahead the informativeness of the virtual instances yields good performance.
>
> | Method| CIFAR-10 |
> |----------|-------------------------------- |
> | Ent w. AA | 53.44 $\pm$ 1.94 |
> | LADA$^{fixed}_{EntAA}$ | 55.22 $\pm$ 0.86 |
>
> However, this aspect has two problems as the below:
>
> - First, AutoAugment requires the labeled dataset, which is not available in the active learning setting, i.e., the learnt AutoAugment comes from the training with the full labeled dataset of CIFAR-10. Therefore, the learnt AutoAugment could have been cheating under the assumption of joint acquisition and augmentation.
>
>
> - Second, the pre-trained AutoAugment does not select the augmentation to maximize the acquisition score of the unlabeled instances, so the acquisition with AutoAugment would not be optimal considering the missing contribution of labeling hard instances. Some hard instances may become very informative after the acquisition because of the augmented virtual instances, but the learnt AutoAugment cannot anticipate this opportunity of information gain because it is pre-trained and static. If we were to train AutoAugment in the acquisition process, AutoAugment will require the labeled instances which could be the acquired ones. However, this training of AutoAugment does not follow the problem setting in the paper because LADA aims to learn the optimal augmentation on the unlabeled instances.
>
>
> 2. Answer to Cons 2
>
> We expanded the baseline of Core-Set to compare it against LADA during the rebuttal period. The results are as the below.
>
> | Method | Fashion | SVHN | CIFAR-10 | CIFAR-100 | Time | Param. |
> | ------------------------| ----------------------- | ------------------------| ------------------------| ------ | ------ | ------ |
> | Core-Set | 83.96 $\pm$ 0.55 | 76.89 $\pm$ 0.50 | 51.45 $\pm$ 0.82 | 43.90 $\pm$ 0.76 | 1.54 | $-$ |
>
> We will add the results in Table 1 of the paper.
>
> 3. Answer to Cons 3
>
> We already presented such experiments in Appendix B.3, and we enumerated results in Appendix Figure 3 and Appendix Table 5 under the same setting of [2] and [3]. The experiments demonstrate the benefit of adopting LADA.
>
>
> 4. Answer to Cons 4
>
> We will strengthen the explanation further in Section 3.3.1 as well as Section 3.3.2. Having said that, here are the basic steps to integrate manifold mixup (a.k.a. InfoMixup in our paper) into LADA.
>
> Eq. 8 calculates the latent representation of the pair of the images at the k$^{th}$ layer. Eq. 9 is the policy generator network specification for InfoMixup. Eq. 10 calculates the N number of the mixed latent representation by $\tau$, which is the output of the generator network. Then, the N number of the mixed latent representations are feed-forwarded until the last layer of the network to calculate the averaged prediction entropy. The averaged prediction entropy of the N number of representations are used in Eq. 11 to get the gradient signal that trains the policy generator network, and the optimal $\tau$ is calculated as Eq. 12. This optimal $\tau$ becomes the augmentation policy of LADA.

---

> > ### Comment · Reviewer_pwAg · 2021-08-13
> > **Core-Set approach**
> >
> > I am really skeptical of the core-set numbers mentioned here. It seems that the core-set approach performs even worse than MaxEntropy and VarRatio based approaches when it shouldn't be the case. Can the authors do the baselining more effectively? The approximation of core-set can be k-centre greedy approach. A standard implementation of it which can be used is - https://github.com/google/active-learning/blob/master/sampling_methods/kcenter_greedy.py
> > Can authors report results with this?

---

> > > ### Author Response · Authors · 2021-08-14
> > > **Response to Core-Set approach**
> > >
> > > 1. The code with which we experimented is provided at https://github.com/JordanAsh/badge/blob/master/query_strategies/core_set.py, which uses k-center greedy approximation. We will conduct the experiment with the code that the reviewer suggested, and report the results soon in this rebuttal period.
> > >
> > >
> > > 2. Even though the original core-set approach performed better than Entropy or VarRatio, the original core-set approach was conducted with a large amount of budget. However, there exists a report where the core-set performs worse than uncertainty, when the amount of budget is small (Towards robust and reproducible active learning using neural networks, Munjal, P., et al, Arxiv 2020). Table 11~15 in this paper compares diverse active learning methods using the CIFAR-100 dataset, and Uncertainy (Entropy) acquisition outperforms Core-set for several trials, depending on the initial labeled set.
> > >
> > >
> > > 3. We conjecture that there exists a trade-off between the diversity-based acquisition and the uncertainty-based acquisition, and the trade-off depends on the decision boundary construction by the classifier. When the decision boundary is not yet solidified, the uncertainty-based acquisition is more effective by selecting the instances near the boundary to solidify the boundary (e.g., exploitation). However, after learning the decision boundary, the diversity-based acquisition is more effective by selecting the representative instances (e.g., exploration). For such reason, in our experiment results in Table 1 of the paper, Core-set outperforms Entropy acquisition for Fashion and SVHN dataset, which are relatively easy for the classifier to solidify the decision boundary.

---

> > > > ### Comment · Reviewer_pwAg · 2021-08-14
> > > > **Follow up discussion**
> > > >
> > > > I thank the authors for a healthy discussion. I will be willing to change my ratings once the authors finish up the above experiments. Experiment with a single dataset (for example - Cifar10) would suffice.
> > > >
> > > > As another point, I believe the authors should also compare with semi-supervised approaches such as MixMatch [1]. Such semi-supervised approaches also rely on heavy augmentations over unlabelled datasets and often achieve much superior performance. What then is the advantage provided by the LADA approach?
> > > > While it's not justifiable to compare active learning approaches with semi-supervised learning approaches but since this paper's contributions relies a lot on augmentations so this discussion is essential for this paper.
> > > >
> > > >
> > > > [1] MixMatch: A Holistic Approach to Semi-Supervised Learning. Berthelot et al.

---

> > > > > ### Author Response · Authors · 2021-08-15
> > > > > **Response to Follow up discussion**
> > > > >
> > > > > 1.
> > > > > As the reviewer pointed out, it is difficult to compare MixMatch and LADA, since MixMatch is a holistic algorithm for semi-supervised learning, while LADA provides acquisition criteria for (supervised) active learning. However, augmentation is indeed an important part of both methods, so we compare them from the perspective of data augmentation as follows.
> > > > >
> > > > > - Augmentation in MixMatch is used to produce a pseudo label of unlabeled instance, while augmentation in LADA is used to search for the informative instance to query the oracle label. This difference comes from the fact that the approach to the ground-truth label is not allowed for semi-supervised learning.
> > > > >
> > > > > - Augmentation in MixMatch is static with fixed policy, while augmentation in LADA is optimized to maximize the acquisition score of the augmented instances, so that the classifier can be trained with maximally informative instances adaptively.
> > > > >
> > > > >
> > > > > 2.
> > > > > Some spirits of MixMatch may be used for active learning, e.g., selecting the instances that have high entropy values when averaging the logits of k augmentation results and sharpening the logits. Having said that, the CAL method (see row 6 of Table 1 in the paper) is such kind of the adoption of the semi-supervised spirit to the active learning, where the variation of logits of k augmentation results becomes the acquisition criterion. As Table 1 shows, LADA outperforms or is comparable to CAL.
> > > > >
> > > > >
> > > > > 3.
> > > > > The experiments of CAL also indicate that MixMatch may be also applicable for active learning, i.e., we can select instances via LADA and then apply MixMatch-based semi-supervised loss for training the classifier. We can devise this experiment case that joins LADA and MixMatch, so this new case can show that both methods are independent approaches and able to boost the performance, jointly. We are more than willing to perform this experiment if the reviewer decides that this will change the value of the submitted paper, significantly.
> > > > >
> > > > > Having said that, it is also common knowledge that the semi-supervised learning and active learning are two different methods, and it would be too much to empirically prove that this possible combination will improve the performance. Surely, using the unlabeled pool in the learning process will increase the existing performance. However, that is not the main message of LADA which claims the importance of dynamically learned augmentation value looking ahead of its acquisition with small budgets.
> > > > >
> > > > > Please let us know the best method to improve the paper from your perspective with MixMatch. We will include the above discussion for sure. Would you suggest an additional experiment with LADA+MixMatch?

---

> > > > > > ### Comment · Reviewer_pwAg · 2021-08-16
> > > > > > **Role of augmentations**
> > > > > >
> > > > > > I thank the authors for their thoughts on combining semi-supervised learning and LADA.
> > > > > > I want to reiterate my thought here that prior works of Active Learning have not been compared to semi-supervised learning approaches because as the authors also mention these are two separate lines of work. However, one major distinguishing factor in my opinion has been the role of augmentations, whereby the active learning approaches have rarely relied on augmentations while devising acquisition functions whereas the semi-supervised learning approaches rely heavily on augmentations.
> > > > > > Thus there have been some works trying to combine the two fields [1].
> > > > > >
> > > > > > I would encourage the authors to thus show an experiment combing the semi-supervised approaches (MixMatch or a simpler method if there is an issue of time) and LADA and show its benefit over other active learning approaches.
> > > > > >
> > > > > > As a reminder, please also update me once the experiment with k-center greedy finishes. I will be willing to change my rating on the basis of this experiment.
> > > > > >
> > > > > > References:
> > > > > > [1] Consistency-based Semi-supervised Active Learning: Towards Minimizing Labeling Cost. Gao et al.

---

> > > > > > > ### Author Response · Authors · 2021-08-17
> > > > > > > **Response with Core-set result**
> > > > > > >
> > > > > > > We conducted the experiment with Core-set on the CIFAR-10 dataset, using the code that the reviewer suggested. We modified the code to adopt the ResNet18 as the backbone classifier with the Adam optimizer. We also set the initial labeled set with 20 instances (flag warmstart_size) and the budget per iteration as 10 instances (flag batch_size).
> > > > > > >
> > > > > > > The result is as follow, which shows a similar performance to the original report.
> > > > > > >
> > > > > > > | Method | CIFAR-10 |
> > > > > > > | ------------------------| ----------------------- |
> > > > > > > | Core-Set | 51.47 $\pm$ 1.26 |
> > > > > > >
> > > > > > >
> > > > > > > Also, we are conducting experiments to combine semi-supervised learning and LADA. The result will be posted soon as the experiment finishes.

---

> > > > > > > > ### Comment · Reviewer_pwAg · 2021-08-17
> > > > > > > > **Thanks for the experiment**
> > > > > > > >
> > > > > > > > I have updated my rating.
> > > > > > > > I will wait for the experiment combining semi-supervised learning and LADA to further change my rating.

---

> > > > > > > > > ### Author Response · Authors · 2021-08-20
> > > > > > > > > **Response with LADA in semi-supervised setting results**
> > > > > > > > >
> > > > > > > > > We conducted the experiments by combining semi-supervised loss with active learning. We selected Entropy-based acquisition as the baseline active learning to compare with LADA. We used the CIFAR-10 dataset.
> > > > > > > > >
> > > > > > > > > We started with 250 labeled dataset and expanded the training set by acquisition as 250-500-1000-2000-4000 (i.e., we doubled the acquisition budget at each iteration), following the prior works of MixMatch [1] or the self-ensembling method [2]. After acquiring instances via active learning, we initialized the network and trained the network for 100 epochs. We adopted a consistency-based semi-supervised learning method, resulting in the loss function for semi-supervised learning with the baseline active learning as follow:
> > > > > > > > >
> > > > > > > > > $ L_{Ent}=\frac{1}{|X_L|}\sum_{(x_{i},y_{i})\in X_{L}}{CE(f_{\theta}(x_i),y_i)} + \frac{1}{|X_U|}\sum_{x_{i}\in X_U}{MSE(f_{\theta}(x_i),f_{\theta}(f_{aug}(x_i)))}, $
> > > > > > > > >
> > > > > > > > > and the loss functions with LADA and LADA$^{fixed}$ as follow:
> > > > > > > > >
> > > > > > > > > $ L_{LADA}=\frac{1}{|X_L|}\sum_{(x_{i},y_{i})\in X_{L}}{CE(f_{\theta}(x_i),y_i)} + \frac{1}{|X_U|}\sum_{x_{i}\in X_U}{MSE(f_{\theta}(x_i),f_{\theta}(f_{aug}(x_i)))} + \frac{1}{|X_M|}\sum_{(x_i,y_i)\in X_M}{CE(f_{\theta}(x_i),y_i)}.$
> > > > > > > > >
> > > > > > > > > Here, $X_L$ denotes the labeled dataset, $X_U$ denotes the unlabeled dataset, and $X_M$ denotes the mixup dataset that were selected by LADA.
> > > > > > > > >
> > > > > > > > > We provide the experiment results below. By this experiment, we observed the following.
> > > > > > > > >
> > > > > > > > > - As expected, semi-supervised learning achieves better performance than supervised learning with the aid of the learning representations of the unlabeled dataset.
> > > > > > > > > - LADA and LADA$_{fixed}$ show better performance than the baseline active learning algorithm in both the supervised and semi-supervised settings.
> > > > > > > > > - The performance gain by looking ahead the informativeness of the virtual instances are larger when we are given the smaller number of the labeled dataset.
> > > > > > > > >
> > > > > > > > >
> > > > > > > > > | # labeled data|| 250| 500| 1000 | 2000 | 4000|
> > > > > > > > > | --- | --- | ------------- | ----- | -------- | ------- | ------- |
> > > > > > > > > | Supervised | Entropy | 32.13 $\pm$ 1.15 | 39.43 $\pm$ 2.43 | 49.23 $\pm$ 1.10 | 63.30 $\pm$ 1.64 | 75.19 $\pm$ 1.17 |
> > > > > > > > > |                  | LADA$^{fixed}_{EntMix}$ | 32.13 $\pm$ 1.15 | 41.49 $\pm$ 0.57 | 51.53 $\pm$ 1.27 | 65.54 $\pm$ 1.38 | 76.14 $\pm$ 1.72 |
> > > > > > > > > |                   | LADA$_{EntMix}$ | 32.13 $\pm$ 1.15 | 42.98 $\pm$ 1.83 | 52.32 $\pm$ 1.13 | 65.75 $\pm$ 1.37 | 75.68 $\pm$ 0.61 |
> > > > > > > > > | Semi-supervised | Entropy | 38.34 $\pm$ 0.17 | 43.12 $\pm$ 1.95 | 54.23 $\pm$ 0.90 | 68.53 $\pm$ 0.95 | 81.15 $\pm$ 1.03 |
> > > > > > > > > |                   | LADA$^{fixed}_{EntMix}$ | 38.34 $\pm$ 0.17 | 47.01 $\pm$ 0.71 | 57.02 $\pm$ 1.63 | 70.83 $\pm$ 2.06 | 82.15 $\pm$ 0.86 |
> > > > > > > > > |                   | LADA$_{EntMix}$ | 38.34 $\pm$ 0.17 | 46.54 $\pm$ 1.75 | 57.43 $\pm$ 1.03 | 72.14 $\pm$ 2.02 | 82.33 $\pm$ 0.18 |
> > > > > > > > >
> > > > > > > > > The authors ask the reviewer to consider a significant increase in the ratings given that now LADA is far more expanded than the original version since the empirical result shows the synergetic merge of active learning and semi-supervised learning.
> > > > > > > > >
> > > > > > > > > Also, we sincerely thank the reviewer for the greatly constructive review comments. We felt the value of the review process from this review thread.
> > > > > > > > >
> > > > > > > > > [1] MixMatch: A Holistic Approach to Semi-Supervised Learning. Berthelot et al.
> > > > > > > > >
> > > > > > > > > [2] Temporal Ensembling for Semi-Supervised Learning, Laine et al.

---

> > > > > > > > > > ### Comment · Reviewer_pwAg · 2021-08-23
> > > > > > > > > > **Confusion regarding the experimental setup**
> > > > > > > > > >
> > > > > > > > > > Thanks for conducting the experiments!!
> > > > > > > > > >
> > > > > > > > > > I am slight confused by the numbers mentioned here. So by adopting a consistency-based approach, do you mean using some variant of MixMatch, and not MiXMatch directly?
> > > > > > > > > >
> > > > > > > > > > The MixMatch numbers are pretty high compared to the numbers mentioned above. Even for the different baselines mentioned in Table 5 in [1], all the numbers are much higher. Can the authors use any of these baselines as the self-supervised method?
> > > > > > > > > >
> > > > > > > > > > [1] MixMatch: A Holistic Approach to Semi-Supervised Learning. Berthelot et al.

---

> > > > > > > > > > > ### Author Response · Authors · 2021-08-25
> > > > > > > > > > > **Response to Confusion regarding the experimental setup**
> > > > > > > > > > >
> > > > > > > > > > > We are sorry for the confusing results and insufficient explanations. Below, we provide a detailed explanation of our semi-supervised experiment, the reason for the low accuracy, and the new experimental plan.
> > > > > > > > > > >
> > > > > > > > > > > 1) Details of our semi-supervised experiment
> > > > > > > > > > >
> > > > > > > > > > > Yes. As the reviewer expected, our experiment is not the integration of LADA+MixMatch. We selected the $\Pi$-model for the semi-supervised learning, because of the time complexity issue. We show the loss function of the $\Pi$-model and the MixMatch model, as the below.
> > > > > > > > > > >
> > > > > > > > > > > $ L_{\Pi -model}=\frac{1}{|X_L|}\sum_{(x_{i},y_{i})\in X_{L}}{CE(f_{\theta}(x_i),y_i)} + \frac{1}{|X_U|}\sum_{x_{i}\in X_U}{MSE(f_{\theta}(x_i),f_{\theta}(f_{aug}(x_i)))} $
> > > > > > > > > > >
> > > > > > > > > > > $ L_{MixMatch}=\frac{1}{|X_{L}'|}\sum_{(x_{i},y_{i})\in X_{L}'}{CE(f_{\theta}(x_i),y_i)} + \frac{1}{|X_{U}'|}\sum_{(x_{i}, q_{i})\in X_{U}'}{MSE(q_{i},f_{\theta}(x_i))} $
> > > > > > > > > > >
> > > > > > > > > > > Here, $X_{L}'$ and $X_{U}'$ are mixed labeled dataset and mixed unlabeled dataset, following line 13-14 in Algorithm 1 of [1]; and $q_i$ is the pseudo label of the unlabeled instance, following line 8 in Algorithm 1 of [1]. The added parts of $\Pi$-model and MixMatch model are the MSE losses, which requires a large number compared to the supervised experiments that we have run. Moreover, given a batch of size B, $|X_{U}|=B$ for the $\Pi$-model, while $|X_{U}'|=B * K$ for MixMatch where $K$ is the number of augmentation. Also, MixMatch uses $B*K$ number of unlabeled instances to calculate the pseudo label. Eventually, the below table enumerates the increased time complexity of MixMatch from $\Pi$-model.
> > > > > > > > > > >
> > > > > > > > > > > |            | Cross-Entropy loss || MSE loss || Calculating pseudo label |
> > > > > > > > > > > | ----------  | --- | --- | --- | --- | --------- |
> > > > > > > > > > > |            | Feed-forward | Back-propagation | Feed-forward | Back-propagation | Feed-forward |
> > > > > > > > > > > | $L_{\Pi -model}$ | B | B | B + B | B | $-$ |
> > > > > > > > > > > | $L_{MixMatch}$ | B | B | B * K | B * K | B * K |
> > > > > > > > > > >
> > > > > > > > > > > Once we empirically measure the time complexity, we get the below wall-clock time results for a single epoch training on TitanX GPU.
> > > > > > > > > > >
> > > > > > > > > > > | $\Pi$-model | $\Pi$-model + LADA | MixMatch | MixMatch + LADA |
> > > > > > > > > > > | -------- | -------- | -------- | -------- |
> > > > > > > > > > > | 83.55 sec | 177.07 sec | 173.41 sec | 261.85 sec |
> > > > > > > > > > >
> > > > > > > > > > > The wall-clock time of MixMatch is about twice longer than $\Pi$-model, since the number of instances that are used for calculating the loss function is different in both models. Because of this time complexity, we choose to utilize the $\Pi$-model to demonstrate the integration of semi-supervised and active learning.
> > > > > > > > > > >
> > > > > > > > > > > 2) Reason for the low accuracy
> > > > > > > > > > >
> > > > > > > > > > > The above result that we posted is the result of the shortened training only to show the performance gain by LADA, not to compare the performances between the fully trained models. We limited our training time to be 100 epochs and used ResNet-18 with Adam optimizer of a learning rate 1e-3 as the backbone. Therefore, we had room to improve our performance. Our point was the advance made by the integration between the semi-supervised and the active learning, and the direct performance comparison after such full training was not our purpose. It should be noted that we confirmed that LADA outperforms the baseline active learning with this shortened training as well as the baseline semi-supervised learning with this shorted training.
> > > > > > > > > > >
> > > > > > > > > > > 3) New experimental plan
> > > > > > > > > > >
> > > > > > > > > > > Since the reviewer intends to compare the fully trained performance between the Semi-Supervised model and the LADA+Semi-Supervised model, we are experimenting with the fully trained setting, i.e., increasing the training epochs. We will update the fully trained results to this thread as the performance with 250, 500, 1000 labeled datasets are calculated.
> > > > > > > > > > >
> > > > > > > > > > >  [1] MixMatch: A Holistic Approach to Semi-Supervised Learning. Berthelot et al.

---

> > > > > > > > > > > ### Author Response · Authors · 2021-08-29
> > > > > > > > > > > **Response to Confusion regarding the experimental setup (part 2)**
> > > > > > > > > > >
> > > > > > > > > > > We provide the $\Pi$-model results of the fully-trained semi-supervised setting as the below.
> > > > > > > > > > >
> > > > > > > > > > > We adopted WideResNet as the backbone, and we set the initial learning rate and the maximal weight for the consistency loss as 3e-03 and 100, respectively. We ramped up the learning rate and the consistency loss weight using the Gaussian ramp-up curve during the first 80 epochs. We elongated the total training epoch to 500 epochs.
> > > > > > > > > > >
> > > > > > > > > > >
> > > > > > > > > > > | # labeled data| 250| 500| 1000 | 2000 | 4000|
> > > > > > > > > > > | ------------------ | ------------- | ----- | -------- | ------- | ------- |
> > > > > > > > > > > | $\Pi$-model + Entropy | 45.47 | 56.40 | 66.09 | 75.46 | 81.61 |
> > > > > > > > > > > | $\Pi$-model + LADA$^{fixed}_{EntMix}$ | 45.47 | 59.51 | 68.30 | 77.77 | 82.86 |
> > > > > > > > > > > | $\Pi$-model + LADA$_{EntMix}$ | 45.47 | 58.94 | 68.92 | 78.54 | 82.97 |

---

> > > > > > > > > > > > ### Comment · Reviewer_pwAg · 2021-08-29
> > > > > > > > > > > > **Thanks for the experiments**
> > > > > > > > > > > >
> > > > > > > > > > > > I thank the author for their experiments. While it doesn't seem that there is a big difference between LADA and LADA^{fixed}, it does seem to have small gains. I have updated my rating accordingly and thank the authors for being actively engaged during the discussion period!

---

### Official Review · Reviewer_qwu7 · 2021-07-15

**Rating:** 6
**Confidence:** 4

**Summary:**

The paper presents a technique for selecting unlabeled instances in an active learning loop while also generating highly informative synthetic data for augmentation. The selection of unlabeled query instances is guided by a predictive acquisition score computed using the augmented data.

**Ethical Concerns:**

Privacy issues in acquiring labeled image data.

**Ethics Review Area:**

["Discrimination / Bias / Fairness Concerns", "Privacy and Security (e.g., consent)"]

**Limitations And Societal Impact:**

Section 6 is very short. The paper could consider adding more details.

**Main Review:**

Main Comments
-------------

1. Lines 48-55 -- The concept introduced here as 'look-ahead' in the context of active learning is actually not new. Similar idea (the expected error reduction framework) was introduced in [1] and has also been discussed in [2, 3] among other works. The proposed technique in the current paper seems to extend that idea to the virtual instances. [1] should be cited in the present work.

2. The proposed technique combines and extends several existing works to end up with a substantially complex system that is (1) mostly restricted to standard image domain and (2) as can be understood from Table 1, has very small practical benefit over baselines.

3. Figure 4: It is not clear why BGADL is shown in the figure when it has already been stated (in lines 285-286) that BGADL has worst performance. As a result, this figure presents a biased plot.

4. Lines 182-183 -- An explanation is required for why the cross-entropy loss related to labeled data was not included/combined in Equation 4 when computing optimal \tau?

5. Lines 234-235 -- What exactly does the paper mean by 'After querying the label of X_S to oracle...'? X_S contains a pair of unlabeled instances. Do we get the labels of all unique unlabeled instances in X_S? Or is the query to oracle for a 'pair' of instances (not sure what that means)?

6. Line 207 -- "(x_i, {x'}_i) \in X_U" should be "(x_i, {x'}_i) \in X_U \times \X_U". Needs to be consistent in all places.

7. Line 249, Table 1 -- The definition of the baseline 'Random' is unclear. Does it mean that for Fashion, SVHN, CIFAR-10 1000 random unlabeled instances were selected (10000 in case of CIFAR-100) to be labeled by the oracle and then the algorithm was trained on them?

8. Figure 4: It would be good to plot the *fraction* of data labeled on the x-axis (including or instead of iteration).


Minor
-----

Line 26 -- 'Beside of active learning...' -> 'Besides active learning...'

Line 86 -- 'Recently, Mixup-based...' -> 'In a recent work, ....'

Line 246-247 -- 'we experiment the various' -> 'we experiment with the various'

Line 248 -- 'we experiment the STN' -> 'we experiment with the STN'



References
----------

[1] N. Roy and A. McCallum. Toward optimal active learning through sampling estimation of error reduction. In Proceedings of the International Conference on Machine Learning (ICML), pages 441–448. Morgan Kaufmann, 2001.

[2] Burr Settles. Active learning literature survey. Technical report, University of Wisconsin-Madison Department of Computer Sciences, 2009.

[3] Ksenia Konyushkova, Raphael Sznitman, Pascal Fua, Learning Active Learning from Data, NeurIPS 2017

Post Rebuttal:

I would like to thank the authors for their detailed response.

I'm increasing my score by one point to marginally above acceptance. IMO, the contribution is somewhat low on novelty.

**Time Spent Reviewing:**

2-3 hours

---

> ### Author Response · Authors · 2021-08-04
> **Response to Reviewer qwu7**
>
> Thanks so much for your constructive reviews. Please see our response below.
>
>
> 1. Answer to Main Comments 1
>
> Thank you for pointing out the related works that we missed. We will cite the previous works including [1, 2, 3] in the Introduction section.
>
> As the reviewer stated, [1, 2, 3] suggested the “look-ahead” without acknowledging the value of augmentation in the “look-ahead” setting. We argued the “Look-Ahead Data Acquisition via augmentation” framework, and we only claim our novelty for look-ahead in conjunction with the augmentation of virtual instances. Moreover, “look-ahead” is a necessary concept in any active learning scheme because the active learning requires an active seeking on high-value data instances which will impact the classifier if they are used in the inference, so this assessment on the impact becomes the “look-ahead” in such active learning concept.
>
>
> 2. Answer to Main Comments 2
>
> LADA is not a complex system because the only added model is the policy generator, $\pi_\phi$. All other models are the standard classifier, the acquisition function, and the augment function, which any modeler requires in performing the task in the active learning settings. The policy network is a simple neural network without a serious structure aspect. The innovation lies on the gradient signal to train the policy network coming from the prediction entropy of Eq. 4.
>
>
> 3. Answer to Main Comments 3
>
> Figure 4 compares the active learning algorithms that use data augmentation, such as CAL and BGADL. These are the only two variants merging the active learning and the data augmentations up to date. Therefore, CAL, BGADAL, and LADA are the only three comparable alternatives to integrate the acquisition and the augmentation, so their performances are compared. Here, LADA shows the best performance in CIFAR-10.
>
>
> 4. Answer to Main Comments 4
>
> As the reviewer suggested, the optimal setting of $\tau$ can be influenced by two factors, which are 1) past labeled instances and 2) current unlabeled instances. However, it may be hard to re-use the past labeled instances for training the policy generator directly because such labeled instances will ever increase, and the policy network needs to be retrained for every acquisition step. Therefore, we utilize the past labeled instances indirectly via the trained $f_\theta$, where $f_\theta$ is trained by the past labeled instances, and its predictions are all influenced by such past labeled data. Then, we utilize the entropy of the prediction in calculating Eq. 4. Additionally, the current unlabeled instances are also utilized in Eq. 4 by becoming the input to the augmentation and the classification.
>
>
> 5. Answer to Main Comments 5
>
> Yes. LADA will receive the labels of all unique unlabeled instances in $X_S$.
>
>
> 6. Answer to Main Comments 6
>
> Thank you for pointing out the inconsistent notation. We will replace the term correctly throughout the paper.
>
>
> 7. Answer to Main Comments 7
>
> Random baseline selects 10 instances for Fashion, SVHN, and CIFAR-10 (100 instances for CIFAR-100) randomly at each acquisition iteration, and the selection is repeated for 100 acquisition iterations. Random baseline does not use any acquisition function that selects instances by informativeness.
>
>
> 8. Answer to Main Comments 8
>
> We will modify Figure 4, so that the x-axis contains the fraction of data labeled.
>
>
> 9. Answer to Minor Comments
>
> Thank you for pointing out the wrong representations. We will correct them according to the reviewer's suggestion.

---

> > ### Comment · Reviewer_qwu7 · 2021-08-24
> > **Post Rebuttal**
> >
> > I would like to thank the authors for their detailed response.
> >
> > I'm increasing my score by one point to marginally above acceptance. IMO, the contribution is somewhat low on novelty.

---

### Official Review · Reviewer_hTM1 · 2021-07-15

**Rating:** 6
**Confidence:** 3

**Summary:**

Summary

The paper’s main goal is to incorporate data augmentation schemes into an active learning framework to further reduce the need for labeled data. An approach is proposed that ensures that the augmentations generated are also informative from the learning perspective. Experimental results are provided that show that the proposed approach is better than a simple baseline where the augmentations are generated for candidate instances without considering their informativeness.


**Limitations And Societal Impact:**

The limitations section is short which mainly says that the approach is limited to classification tasks. Any discussion on failure modes or sensitivities to architectural choices, etc will be useful. It is not clear what the authors mean by "privacy issue is relevant". This needs a bit more clarification.

**Main Review:**

Originality

The idea of looking ahead and considering the informativeness of the augmentations for the data acquisition process is interesting and useful. The main effect of the joint approach is that selected instances and associated augmentation are on average more informative. As the authors point out, a similar idea has been considered in [19]. The key difference being that the augmentation scheme is learned instead of being fixed as in [19].

Significance

Improving the data efficiency for the active learning process is an important problem. Optimizing the data augmentation scheme during the data acquisition process seems effective in reducing the need for labeled data.

Quality

The paper is well motivated and easy to read. Empirical comparisons show that the method does improve over BGADL and CAL baselines. Qualitative results suggest that the approach chooses different instances compared to MaxEntropy scheme and considers the space of augmentations as well when choosing them.

Other questions/Comments
InfoMixup seems to perform better than InfoSTL, is the only reason that STL can only generate labels preserving in the augmentations? Is this also the only reason for improvement over CAL or is it the richer kinds of augmentations enabled by manifold mixup? Isolating the gains will be helpful.


**Time Spent Reviewing:**

4

---

> ### Author Response · Authors · 2021-08-04
> **Response to Reviewer hTM1**
>
> Thanks so much for your constructive reviews. Please see our response below.
>
>
> 1. About the performance of STN
>
> As the reviewer stated, STN is label-preserving augmentation, and CAL also adopts label-preserving augmentations such as random cropping and flipping. Hence, it lacks exploring latent space by generating diverse virtual instances, and it leads to a small amount of performance improvement compared to Mixup augmentation.
>
>
> 2. About the limitations and societal impact
>
> We will strengthen the Limitation section. Also, the privacy issue might happen when collecting and labeling data instances.

---

### Official Review · Reviewer_9dkm · 2021-08-01

**Rating:** 7
**Confidence:** 5

**Summary:**

This paper proposes an interesting alternative approach to active learning. The main idea is an interesting combination of active learning and data augmentation by selecting instances that are informative with and without augmentation. Moreover, the proposed approach allows to optime the data augmentation policy instead of using a fixed policy by maximizing the acquisition score. Experiments shows that the proposed approach improves previous approaches, and it can be combined with different acquisition approaches.

The main contributions of the paper are the following:

• An interesting way to combine both active learning and data augmentation instead of a straightforward pipeline.

• The idea of using the large unlabeled dataset to optimize the augmentation policy allows to learn better data augmentation needed for further acquisition steps.


**Ethical Concerns:**

no ethical concerns

**Ethics Review Area:**

["I don’t know"]

**Limitations And Societal Impact:**

no limitations

**Main Review:**

# Originality

As it is mentioned in the previous section, the paper proposes an interesting idea of selecting samples that are informative with and without augmentation. Moreover, the augmentation policy is learnt using all unlabeled pool of instances instead of using the labelled pool; hence maximizing the predictive acquisition score. The augmentation consists of introducing different data augmentations: (a) a Spatial Transformer Network (STN) that transforms an instance using a grid sampler, (b) Mixup, (c) Manifold Mixup and (d) AdaMixup. These transformations are interesting to modify the instance.

Question 1: Instead of learning the augmentation policies that creates new virtual instances, it will be interesting to learn policies on traditional augmentation techniques like cropping, flipping, rotation and the transformation strength. Moreover, there are differentiable techniques and fast algorithms to train such policies. Therefore, this comparison will be interesting to improve the quality and significance of the paper and allows the comparison with active learning paper that selects instances that are consistent to the augmentations.

# Quality

The paper shows many experiments comparing the proposed approach to other methods; showing that the proposed approach is comparable to and better to the state of the art. However, the presentation of these results could be improved since several augmentation techniques are mentioned, but few of them are only shown in the entropy-based methods and not shown in other acquisition functions.

Question 2: Is the proposed approach only able to learn Mixup augmentation policies for all the acquisition functions, but not the other augmentations? If so, why is it only shown these cases. The paper should clarify this approach in order to validate the usefulness of the idea.

Question 3: Section B.2 of the supplemental material shows the \tau is similar across acquisition iterations; therefore, it shows that learning once will be enough during the time or setting it by cross-validation. How could it affect the value of tau to the proposed approach? And also its benefits? A correlation between tau and the performance will be interesting to show. How are the values of the other augmentation techniques?


# Significance

The paper proposes an interesting and different approach to combine data augmentation and active learning, and benefit the community to join the efforts like semi-supervised learning with the consistency-learning. However, as mentioned before, the paper should answer the previous questions to highly impact the community and validate the proposed approach.

# Clarity

The clarity of the paper is the main weakness, because there is a lot of information in the paper like acquisition functions, augmentations, procedures, gradients, etc that difficult the understanding of the paper. Presenting a main idea like Mixup augmentation together with the acquisition score and active learning and then a section with all possible variants, or a general and understandable methodology and notation will be improve the quality of the paper.

Therefore, the paper shows an interesting idea, but needs to answer previous questions before being accepted in order to improve the significance and quality of the paper.


**Time Spent Reviewing:**

5

---

> ### Author Response · Authors · 2021-08-04
> **Response to Reviewer 9dkm**
>
> Thanks so much for your constructive reviews. Please see our response below.
>
>
> 1. Answer to Question 1
>
> STN, with which we conducted the experiments, is an augmentation of spatially transforming the data instance, which is similar to rotation, flipping, etc. Therefore, we think that STN could be the conventional augmentation method, and LADA shows some benefit by incorporating STN in the acquisition process, see rows 12-13 of Table 1 in the paper.
>
> In addition, AutoAugment [1] corresponds to the algorithm that learns policies on such conventional augmentation techniques. AutoAugment designs multiple policies with fixed augmentation hyper-parameters, and selects the optimal policy, while LADA learns the hyper-parameters of the policies by the generator network. Since LADA is a generalizable framework, LADA is possible to adopt AutoAugment as an augmentation function, if we construct the policy generator network of LADA to output the selection of the augmentation of flip, rotation, etc, and their required augmentation parameters, as well. This adoption still consistently maintains the learning direction of maximizing the entropy of the virtual instances as stated in Eq. 4, and still is an instantiation of the LADA framework.
>
>
> 2. Answer to Question 2
>
> Other augmentations such as STN is applicable for all the acquisition functions. When applying STN and Mixup to the Entropy acquisition function, we confirmed that Mixup shows higher improvement, see rows 10-11 and rows 12-13 of Table 1 in the paper. Hence, we applied Mixup to other acquisition functions.
>
> During the rebuttal period, we conducted the experiments by applying STN to other acquisition functions. The results are as the below.
>
> | Method || Fashion| SVHN | CIFAR-10      | CIFAR-100      |
> | --- | --- | ------------- | ----- | -------- | ------- |
> | VarRatio-based | LADA$^{fixed}_{VarSTN}$ | 83.32 $\pm$ 0.75 | 74.70 $\pm$ 0.75 | 54.22 $\pm$ 0.91 | 46.07 $\pm$ 0.31 |
> |                   | LADA$_{VarSTN}$ | 83.35 $\pm$ 0.56 | 74.86 $\pm$ 1.53 | 55.76 $\pm$ 0.53 | 46.42 $\pm$ 0.40 |
> | VAAL-based | LADA$^{fixed}_{VAALSTN}$ | 83.32 $\pm$ 0.77 | 72.86 $\pm$ 1.59 | 51.33 $\pm$ 0.13 | 44.27 $\pm$ 0.26 |
> |                   | LADA$_{VAALSTN}$ | 83.56 $\pm$ 0.53 | 74.53 $\pm$ 1.65 | 53.78 $\pm$ 2.24 | 45.06 $\pm$ 1.29 |
> | LL4AL-based | LADA$^{fixed}_{LL4ALSTN}$ | 83.69 $\pm$ 0.28 | 74.63 $\pm$ 1.82 | 53.28 $\pm$ 0.67 | 45.01 $\pm$ 0.90 |
> |                   | LADA$_{LL4ALSTN}$ | 83.16 $\pm$ 0.22 | 74.74 $\pm$ 1.17 | 53.17 $\pm$ 0.22 | 45.94 $\pm$ 0.61 |
>
> We will add these results to the Table 1 and Figure 4 in the paper.
>
>
> 3. Answer to Question 3
>
> As Figure 2 of Appendix shows, the value of $\tau$ is relatively small at the initial iterations. In the initial iterations, the model is not trained well, so the vicinal virtual instances, which are generated from a small value of $\tau$, are informative. However, after some iterations of learning the classifier, we need to explore a broader data space to generate informative virtual instances. This leads to the increased value of $\tau$. Hence, learning $\tau$ dynamically at each iteration is related to the performance of the current classifier, and Table 1 in the paper shows that this dynamic learning of $\tau$ increases the performance over the static $\tau$ which is set once.
>
>
> [1] AutoAugment: Learning Augmentation Policies from Data. Cubuk et al.

---

> > ### Comment · Reviewer_9dkm · 2021-09-02
> > **response to author**
> >
> > After reading all the reviews and author feedback, I upgraded my score and recommend the acceptance of the paper because it shows benefits and could help the community by combining augmentation+active learning and semi supervised learning.

---

### Author Response · Authors · 2021-08-10
**Thank you for the constructive reviews**

We really appreciate the reviewers for the constructive reviews.

We would like to emphasize that the novelty of LADA lies in the idea that it looks ahead the informativeness of the virtual instances that will be generated from the unlabeled instances. Furthermore, to generate maximally informative virtual instances, LADA learns the augmentation policy with the objective of maximizing the acquisition scores of the virtual instances generated from unlabeled instances.

Having said that, the reviewers pointed out that the AutoAugment [1], which learns the augmentation policy from the data, should be also compared to our framework. However, there exist some difficulties in applying AutoAugmnet to our LADA framework for the reasons below.

First, AutoAugment requires a labeled dataset to learn the optimal augmentation policy, which is unavailable in the acquisition phase of the active learning setting. It may be possible to learn AutoAugment once and use it as the optimal policy (a.k.a. LADA$^{fixed}$), but this could be considered as cheating in the perspective of active learning.

Second, AutoAugment learns the optimal augmentation policy with the whole labeled train set. However, the proposed LADA aims at optimizing the augmentation policy to maximize the informativeness of the virtual instances that will be generated from the unlabeled instances, with the feedback of the current classifier. Hence, the learnt AutoAugment may not ensure the informativeness of the virtual instances for the current classifier.

However, AutoAugment indeed provides meaningful inspiration by combining the selection among candidates of the augmentation techniques and the parameter calibration of such techniques. Since LADA is a generalizable framework, LADA is possible to adopt AutoAugment as the augmentation component, by constructing the policy generator network of LADA to output the selection of the augmentation of flip, rotation, etc, and their required augmentation parameters, as well. This adoption still consistently maintains the learning direction of maximizing the informativeness of the virtual instances (e.g., predictive entropy as stated in Eq. 4), and still is an instantiation of the LADA framework.

[1] AutoAugment: Learning Augmentation Policies from Data. Cubuk et al.

---

### Decision · Program_Chairs · 2021-09-27

**Decision:**

Accept (Poster)

**Comment:**

The manuscript proposes to combine data augmentation with active learning. Instead of simply applying augmentation on top of the data selected by the acquisition function, the manuscript aims to also involve data augmentation in the data acquisition step. Data augmentation policy is optimized to increase the acquisition function score, and the resulting augmented data is then used in the search process.

Reviewers agreed that the idea of looking ahead and considering the informativeness of the augmentations for the data acquisition process is interesting and useful. As the authors point out, a similar idea has been considered; the key difference being that the augmentation scheme is learned instead of being fixed. Reviewer qwu7 also pointed out other related works on 'look-ahead' in the context of active learning. The authors emphasised that the novelty of the work lies in the look-ahead in conjunction with the augmentation of virtual instances. During rebuttal, Reviewer pwAg guided authors to produce stronger baselines, and synergetic merge of active learning and semi-supervised learning.